# HIERARCHICAL INSTRUCTION-AWARE EMBODIED VISUAL TRACKING

## ABSTRACT

User-centric embodied visual tracking (UC-EVT) requires embodied agents to follow dynamic, natural language instructions specifying not only which target to track, but also how to track—including distance, angle, and directional constraints. This dual requirement for robust language understanding and low-latency control poses significant challenges, as current approaches using end-to-end RL, VLM/VLA, and LLM-based methods fail to adequately balance comprehension with low-latency tracking. In this paper, we introduce **Hierarchical Instruction-aware Embodied Visual Tracking (HIEVT)**, which decomposes the problem into on-demand instruction understanding with spatial goal generation (high-level) and asynchronous continuous goal-conditioned control execution (low-level). HIEVT employs an *LLM-based Semantic-Spatial Goal Aligner* to parse diverse human instructions into spatial goals that directly specify desired target positioning, coupled with an *RL-based Adaptive Goal-Aligned Policy* that enables real-time target positioning according to generated spatial goals. We establish a comprehensive UC-EVT benchmark using over 1.7 million training trajectories, evaluating performance across one seen environment and nine challenging unseen environments. Extensive experiments and real-world deployments demonstrate HIEVT's superior robustness, generalizability, and long-horizon tracking capabilities across diverse environments, varying target dynamics, and complex instruction combinations. The complete project is available at `https://sites.google.com/view/hievt`.

## 1 INTRODUCTION

Providing Human-Computer Interaction has becoming increasingly appleaing for for modern intelligent robots and downstream applications (Zuo et al., 2025; Olaiya et al., 2025; Hoffman et al., 2024; Wu et al., 2025; Pueyo et al., 2024). We introduce **User-Centric Embodied Visual Tracking (UC-EVT)** task, an extension of the previous Embodied Visual Tracking Task (EVT), as users demand more dynamic and interactive systems beyond fixed targets and distance (Van Toan et al., 2023; Zhang et al., 2023). In particular, users expect systems that can respond effectively in dynamic and complex scenarios (Li et al., 2023; Zhou et al., 2024; Ma et al., 2023), quickly comprehend instructions and adapt to new assigned targets, varying in appearance, speed and expected angle of view, without restarting to modify tracking parameters or switch targets. Motivated by the above needs, this paper proposes three core requirements for UC-EVT: **1) User Instructions Understanding.** The UC-EVT agent must be capable of interpreting user instructions, such as natural language commands, to dynamically adjust the tracking agent, including targets, angles, and distances. **2) Real-Time Responsiveness.** The agent must operate in real time, maintaining robust performance while receiving continuous instructions or switching to different targets in dynamic environments. **3) Flexibility and

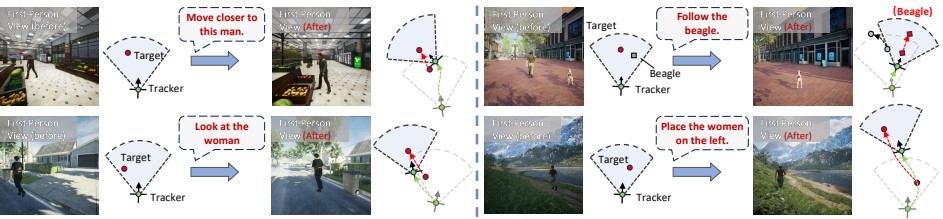

Figure 1: Examples of User-Centric embodied visual tracking with diverse instructions.

**Generalizability.** The agent should flexibly adapt to targets with various appearances and speeds, unseen scenarios, diverse instructions, and evolving spatial relationships, without retraining.

Existing methods (Luo et al., 2019; Zhong et al., 2019b; 2021; 2023; Wang et al., 2025) primarily focus on **Embodied Visual Tracking (EVT)** task, aiming to maintain a stable distances or angles relative to a target by adjust its tracking behaviors. Reinforcement learning (RL)-based models, for example, utilize the distance from the target to the expected location (usually at the center of the view) as the reward to train tracking policy in an end-to-end manner (Luo et al., 2019; Zhong et al., 2019b; 2021). Further, reasearchers leverage vision foundation models (Zhong et al., 2024) to faciliate the domain gap. Recently, systems like Vision-Language-Action Models (VLA) (Kim et al., 2024; Wang et al., 2025) and general-purpose large models (e.g., LLMs and VLMs) have demonstrate the capablity on EVT task while maintain the comprehension ability at same time.

Despite these advancements, existing methods face significant limitations when applied to UC-EVT tasks: **1) Limited Comprehension and Flexibility in RL-based EVT:** RL-based EVT models struggle to handle complex user instructions and exhibit poor transferability across different environments and embodiments. Their rigid tracking strategies (e.g., fixed distances or angles) hinder adaptability to user-centric instructions. **2) Limited Generalization of VLA Models:** VLA models rely heavily on large-scale annotated data, making them ill-suited for unseen conditions such as novel environments, instructions, targets, or agents. **3) Inference Latency in Large Models:** While large models exhibit strong capabilities, their inference speeds (typically 0.5–3 FPS) are insufficient for real-time tracking, resulting in frequent target loss during fast movements.

To address these limitations, we propose a **Hierarchical Instruction-aware Embodied Visual Tracking (HIEVT)** agent that integrates instruction comprehension models with adaptive tracking policies and introduces intermediate spatial goals to bridge human instructions and agent behavior. Specifically, we introduce an *LLM-based Semantic-Spatial Goal Aligner*, serving as a high-level planning module, translating diverse instructions into explicit spatial goals, complement for the insuffciency of comprehension while maintaining spatial precision. The low-level *RL-Based Adaptive Goal-Aligned Policy* modeling temproal spatial information and explict spatial goals at same image space, enhancing the agent's generalization on diverse target speed while preserve a precise tracking approximate to the expected spatial goal. Meanwhile, the low-level policy integrate the vision foundation models (VFMs) to faciliate task irrelvant visual features, ensuring the learned policy is category-agnostic. Our hierarchical design follows a asynchronous updating mechanism, the low-level policy continuously tracking given the latest goal generated from high-level module without blocking on LLM/VLM inference, ensuring a balance of semantic reasoning and real-time performance.

Our contributions are summarized as follows: 1) We introduce the User-Centric Embodied Visual Tracking (UC-EVT) task, which lays the foundation for user-centric human-robot interactions, enabling robots to follow users and provide personalized services. 2) We propose a novel Hierarchical Instruction-aware EVT (HIEVT) model that effectively addresses the limitations of state-of-the-art (SOTA) models while preserving their advantages. 3)We establish a comprehensive UC-EVT benchmark by extending public EVT benchmark with instruction-based tracking, variable target speeds, 10 diverse environments, over 1.7M annotated trajectories, and competitive baselines including classical control, RL policies, and SOTA VLA/VLM models. 4) Our comprehensive experiments evaluate the HIEVT's superiority in terms of general tracking performance, dynamic instruction adaptation, generalization ability, and real-time responsiveness in both simulation and real-world transferability.

## 2 RELATED WORKS

**Embodied Visual Tracking (EVT)** is a foundational skill of embodied AI. Early EVT systems (Yoshimi et al., 2006; Ye et al., 2023; Müller & Koltun, 2021) often relied on passive visual feature extractors combined with handcrafted controllers such as PID or Kalman filters to drive the robot toward the target. While these approaches achieved basic tracking functionality, they lacked robustness in cluttered or dynamic environments and could not flexibly adapt to different tracking objectives or user demands. To improve robustness and generalisation in complex environments, recent EVT research has shifted toward reinforcement learning (RL)-based method (Zhong et al., 2019a; 2021). Recent work Zhong et al. (2024) combines the visual foundation model (Cheng et al., 2023a) and offline reinforcement learning to improve the training efficiency and generalization of the tracker. However, these methods train the agent to track at a specific relative position to the target. If we need

to change the goal, fine-tuning the policy network is required to adapt to new goals. This limits the flexibility and applicability of the agents across varied scenarios and tasks.

**Instruction-aware Robot** is developed to complete tasks by understanding and following human instructions, bridging the gap between high-level human intentions and low-level robotic actions. Early works such as Touchdown (Chen et al., 2019) and R2R (Anderson et al., 2018) focused on vision-language navigation in real-world environments. Subsequent efforts leveraged large pre-trained models to fuse multimodal instructions and enhance generalization. For instance, PALM-E (Driess et al., 2023) builds on a large language model for embodied reasoning, AVLEN (Paul et al., 2022) incorporates audio and natural language in 3D navigation, and VIMA (Jiang et al., 2022) uses multimodal prompts for systematic generalization. While recent VLA paradigm like TrackVLA (Wang et al., 2025)shows that VLAs can be extended to EVT, it still exhibits limited generalization to unseen target categories and various dynamics. Furthermore, these models cannot accurately follow fine-grained user instructions that require precise spatial placement of the target, which is essential in UC-EVT. These limitations highlight the need for a hierarchical framework that integrates the reasoning strength of large models with the real-time responsiveness of lightweight control policies.

## 3 Hierarchical Instruction-aware Embodied Visual Tracking

In this section, we present the design insight of our proposed model, **Hierarchical Instruction-aware Embodied Visual Tracking (HIEVT)** . The core challenge of the User-Centric Embodied Visual Tracking (UC-EVT) is the giant gap from the user instruction $\mathcal{I}_t$ and the actual state $\mathcal{S}_t$ of the tracker. Moreover, the instruction $\mathcal{I}_t$ is given by a user-friendly mode rather than an agent-friendly mode. To bridge the gap, it is necessary to import an intermediate goal $\mathcal{G}_{inter}(t)$ to bridge the user's instruction and the agent's state. This decomposition allows for user instruction understanding as well as efficient agent decision-making, formulated as:

$$\mathcal{D}(\mathcal{I}_t, \mathcal{S}_t) \approx \underbrace{\mathcal{D}(\mathcal{I}_t, \mathcal{G}_{inter}(t))}_{\text{Semantic-Spatial Goal Aligner}} + \underbrace{\mathcal{D}(\mathcal{G}_{inter}(t), \mathcal{S}_t)}_{\text{Adaptive Goal-Aligned Policy}} , \qquad (1)$$

where $\mathcal{D}(\mathcal{I}_t, \mathcal{G}_{inter}(t))$ represents the distance between the user instruction and the intermediate goal, and $\mathcal{D}(\mathcal{G}_{inter}(t), \mathcal{S}_t))$ represents the distance between the intermediate goal and the agent state. We then detail the design of the key components, LLM-based Semantic-Spatial Goal Aligner and RL-based Adaptive Goal-Aligned Policy, the entire model structure is illustrated in Figure 2. At inference time, our system adopts **asynchronous processing** between goal generation and policy execution. When the Semantic-Spatial Goal Aligner receives a new instruction, it may incur variable latency due to LLM inference or communication delays. Rather than blocking the control loop, the system maintains a continuous and stable tracking by using the recent available spatial goal $\mathcal{G}_*$. This design ensures the adaptive goal-aligned policy runs smoothly without pausing(up to 50 fps with lightweight VFM configuration), while the goal aligner operates at its own pace. This asynchronous architecture decouples slow semantic reasoning from fast control, enabling both rich instruction understanding and real-time control.

### 3.1 LLM-based Semantic-Spatial Goal Aligner

The Semantic-Spatial Goal Aligner (SSGA) is the fundamental component responsible for translating diverse user instructions $\mathcal{I}_t$ to a spatial goal $\mathcal{G}_{inter}(t)$. Given the input user instruction $\mathcal{I}_t$, the SSGA outputs an intermediate goal $\mathcal{G}_{inter}(t) = (\mathcal{C}_t, G_t)$, where $\mathcal{C}_t$ is the target category, and $G_t^* = [x_t, y_t, w_t, h_t]$ is a spatial goal, representing the expected target's spatial position in the bounding box format. The SSGA consists of three core components:

**Semantic Parsing** The first step in SSGA is to interpret the user instruction $\mathcal{I}_t$ and extract critical information for goal specification: the target category $\mathcal{C}_t$. This process is performed by the semantic parser $\mathcal{C}_t = \mathcal{P}_{sem}(\mathcal{I}_t)$, where $\mathcal{C}_t \in \mathcal{T}$ indicates the target attributes. For instance, given the instruction "Get closer to the blue car" the semantic parser identifies $\mathcal{C}_t = \{$"blue car"$\}$.

**Spatial-Goal Generation** To resolve ambiguities in user instructions, our parser grounds its interpretation of input instruction $\mathcal{I}_t$ in both linguistic context and visual features. This dual-grounding approach enables robust semantic alignment between natural language commands and environmental

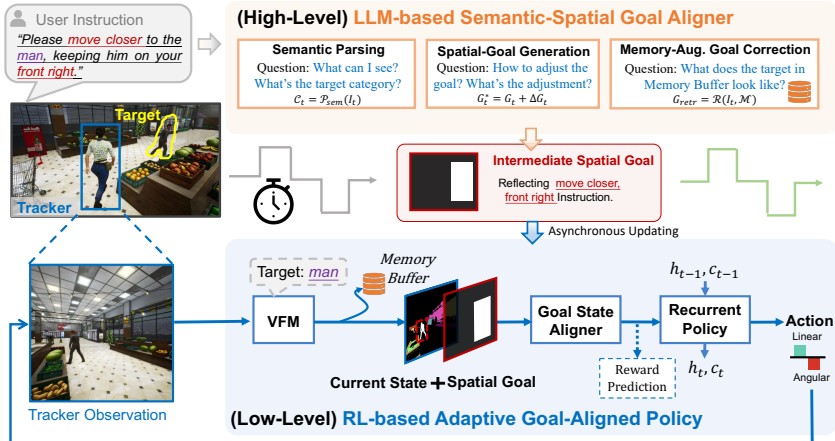

Figure 2: Overview of the Hierarchical Instruction-aware Embodied Visual Tracker (HIEVT). Given a natural language instruction and environmental observation, our system first processes the instruction through the LLM-based Semantic-Spatial Goal Aligner including Semantic Parsing, Spatial-Goal Generation, and Memory-Augmented Goal Correction. This produces a target attribute and a bounding box format spatial goal. The RL-based Adaptive Goal-Aligned Policy then combines this goal with the Visual Foundation Model (VFM) processed observation, feeds them into the following policy network. The Goal State Aligner and Recurrent Policy then generate appropriate action signals to maintain the desired spatial relationship with the target.

observations. Through this process, the parser generates a spatial goal representation $G_t^*$ that accurately reflects the intended spatial directives. This is achieved using a chain-of-thought (COT)-based reasoning mechanism Wei et al. (2022), which incrementally adjusts the current target's spatial representation $G_t$. The spatial goal generation process is formulated as:

$$G_t^* = G_t + \Delta G_t, G_t = \mathcal{B}(\mathcal{VFM}(\mathcal{O}_t, \mathcal{C}_t)) \tag{2}$$

where $\mathcal{VFM}(\mathcal{O}_t, \mathcal{C}_t)$ uses Vision Foundation Models (VFMs) to extract text-conditioned segmentation mask with target-highlighted format given the target category name Zhong et al. (2024), $\mathcal{B}(\cdot)$ detects target's bounding box from segmentation mask by filtering the target's corresponding mask color (white color), $\Delta G_t = [\Delta x_t, \Delta y_t, \Delta w_t, \Delta h_t]$ represents the adjustment to the current target's spatial representation. The adjustment is determined by $\Delta G_t = \mathcal{F}_{COT}(\mathcal{I}_t, G_t)$, where $\mathcal{F}_{COT}$ represents a chain-of-thought reasoning process implemented through a large language models. This reasoning process interprets how the spatial directive in instruction $\mathcal{I}_t$ should modify the current observed target's spatial representation $G_t$, producing adjustment parameters $\Delta G_t$ for both position and size. To guide this interpretation, we developed a system prompt that structures the model's reasoning, ensuring it correctly analyzes spatial relationships (provided in supplementary materials) and translates natural language instructions into geometric transformations. The COT-based approach ensures that the bounding box adjustments are interpretable and aligned with the user's intent, allowing for dynamic and context-aware reasoning about the spatial relationship.

**Memory-Augmented Goal Correction** After generating the spatial goal $G_t^*$, the SSGA applies a correction step using a memory-augmented generation (RAG) mechanism. This step ensures that the generated bounding box is consistent with trajectory priors stored in a memory buffer $\mathcal{M}$. The RAG module retrieves similar instructions and their corresponding bounding boxes from $\mathcal{M}$, i.e., $G_{retr} = \mathcal{R}(\mathcal{I}_t, \mathcal{M})$, where $\mathcal{R}$ is the retrieval function (i.e. Cosine similarity). The retrieved spatial goal $G_{retr}$ is then used for determining if the generated goal $G_t^*$ satisfies the target's physical constraints (e.g., aspect ratio or perspective effect). If conflicts arise, specifically, the Intersection over Union (IoU) (Leal-Taixé et al., 2015; Cordts et al., 2016) value is lower than a threshold (Appendix F.2), the historical mask is used as the final spatial goal $G_t^*$:

$$G_t^* = \begin{cases} G_t^*, & \text{IoU}(G_t^*, G_{retr}) > 0.5 \\ G_{retr}, & \text{IoU}(G_t^*, G_{retr}) \leqslant 0.5 \end{cases} \tag{3}$$

This correction mechanism combines the adaptability of COT-based reasoning with the consistency of target physical constraints, ensuring robust and accurate bounding box predictions.

## 3.2 RL-BASED ADAPTIVE GOAL-ALIGNED POLICY

The proposed Adaptive Goal-Aligned Policy (AGAP) module bridges the intermediate goal $\mathcal{G}_{inter}(t)$ and the user implied state $\mathcal{S}_t$, ensuring precise alignment of the agent's actions with user instructions. This alignment is achieved by an adaptive motion policy which dynamically adjust the agent's movement based on the observation $\mathcal{O}_t$ towards the spatial position indicated by $\mathcal{G}_{inter}(t)$. We leverage VFM-generated mask, to generalize across diverse target objects, as the mask-based representation mitigates domain gaps in texture and appearance variations. The adaptive policy is optimized using offline reinforcement learning with an auxiliary reward regression task to facilitate the alignment, formulated as: $\pi(a_t \mid G^*, \mathcal{O}_t)$. Below, we elaborate on the key components of this module.

**Architechture** The policy consists of two main components: 1) Goal-State Aligner contains multiple convolutional neural network (CNN) layers, serving as the feature extraction and alignment module. It encodes VFM-extracted binary masks highlighting target regions and the spatial goal representation at image-level space, where the mask-based encoding provides appearance-agnostic features that enable generalization to unseen target categories. This aligner outputs a latent aligned representation. Then, a reward prediction layer follows the aligner, serving as the auxiliary task to improve the aligned ability in a self-supervised manner. 2) Recurrent Policy Network consists of a long short-term memory (LSTM) network that models the temporal dynamics of the tracking process to enhance the spatial-temporal consistency of the representation and an actor network $V_\phi$. Moreover, the LSTM's temporal modeling capability enables speed generalization by learning motion patterns across varying target velocities during training. The latent representation from the Goal-state aligner was fed to the LSTM network, followed by the Actor Network to generate motion control action $a_t$. More details about the networks are in supplementary materials.

**Goal-conditioned Offline Policy Optimization** For Adaptive Goal-Aligned Policy (AGAP), our aim is to train a policy that can adapt to diverse spatial goals and achieve precise and fast alignment. Traditional Online RL methods are typically designed for a single-goal objective, and require a huge amount of trial-and-error to converge. Therefore, we use the offline reinforcement learning (Offline-RL) paradigm to train the goal-conditioned policy, considering the training efficiency and dataset diversity's contribution to the overall performance. Meanwhile, we introduce an auxiliary regression task to improve the alignment ability. Below, we detail the reward design, offline-RL training objectives and the auxiliary regression task.

*Training Data Preparation.* We extend the data collection procedure in Zhong et al. (2024) to incorporate a broader range of goal conditions, as the generalization capabilities of our framework critically depend on trajectory diversity. Our dataset $\mathcal{D}$ consists of trajectories $\mathcal{T}_t = (\mathcal{S}_t, \mathcal{O}_t, a_t, r_t, \mathcal{O}_{t+1}, \mathcal{S}_{t+1}, G_t^{final})$, where $t$ is the time step, $\mathcal{S}_t$ and $\mathcal{O}_t$ represent the tracker's state and observation, $a_t$ and $r_t$ denote the action and IoU-based reward, and $G_t^{final}$ specifies the spatial goal. For each episode, we randomly sample goals within the tracker's field of view to ensure diverse spatial configurations, with relative distances $\rho_t \in (200, \rho_{\max})$ and angles $\theta_t \in (-\theta_{\max}/2, \theta_{\max}/2)$. A state-based PID controller with injected noise perturbations generates the goal-conditioned tracking trajectories, enhancing variability and robustness. The final training dataset comprises 10 million steps.

*IoU-based Training Reward.* The reward function $r_t$ is designed to guide the policy move toward aligning $\mathcal{O}_t$ with $G_t^{final}$. At each time step $t$, the reward is defined as:

$$r_t = \text{IoU}(G_t^{final}, \mathcal{B}(\mathcal{VFM}(\mathcal{O}_t, \mathcal{C}_t))), \tag{4}$$

where higher values of Intersection over Union (IoU) indicate the agent is moving towards better alignment in the 2D image.

*Offline Reinforcement Learning.* In this paper, inspired by previous works (Zhong et al., 2024), we extend the standard offline RL algorithms, Conservative Q-Learning (CQL) (Kumar et al., 2020), adapting to goal-conditioned setting. Specificallu, we use two critic networks $Q_\theta^1, Q_\theta^2$ to estimate goal-conditioned Q values. The Q-functions are updated by minimizing the following objective: $L_\theta = \sum_{\mathcal{G}=1}^{\mathbf{N}} L_\mathcal{G}$, where the objective consists of the sum of the losses from all sampled goals in the

batch, with each goal's loss computed according to the following formula:

$$L_{\mathcal{G}} = \mathbb{E}_s \left[ \log \sum_a \exp Q_\theta^i(s, a) - \mathbb{E}_{a \sim \pi_\phi(a|s)}[Q_\theta^i(s, a)] \right]$$
$$+ \frac{1}{2} \mathbb{E}_{s,a,s'} \left[ \left( Q_\theta^i(s, a) - \left( r + \gamma \mathbb{E}_{a' \sim \pi_\phi} \left[ Q_{min}(s', a') - \alpha \log \pi_\phi(a' \mid s') \right] \right) \right)^2 \right] \quad (5)$$

where $\theta$ and $\phi$ are network parameters, $\alpha$ is the entropy regularization coefficient which controls the degree of exploration. $i \in \{1, 2\}$, $Q_{min} = min_{i \in \{1,2\}} Q_\theta^i$, $\gamma$ is the discount factor, $\pi_\phi$ is the learned policy that derived from the actor network $V_\phi$. Note that the state-goal aligner and the recurrent policy are jointly optimized by the RL loss.

*Auxiliary Reward Regression.* For the goal-state aligner, we introduce an auxiliary reward regression task during training to facilitate the alignment between the goal and state representations. This task encourages the agent to recognize high-reward states associated with different goals, effectively steering the learned action policy toward these states. Specifically, we incorporate a fully connected layer following the goal-state aligner, predicting the alignment reward $\hat{r}_t$, the alignment loss is computed using $L_{reg} = \text{MSE}(r_t, \hat{r}_t)$, where MSE denotes the mean squared error, the gradients are only updated to the goal-state aligner. Note that, the output of the fully connected layer will not be fed to the recurrent policy network, which will not affect inference stage.

## 4 EXPERIMENT

In this section, we conduct comprehensive experiments in virtual environments and real-world scenarios, aiming to address the following **five** questions: Q1) Can HIEVT outperform state-of-the-art models on tracking performance, robustness, and generalizability? Q2) How does the adjusting precision and efficiency of HIEVT when there is a new instruction? Q3) How do HIEVT and baselines perform under different target moving speeds and dynamic target category switches? Q4) How do key components of HIEVT affect its performance? Q5) How does HIEVT perform in real world?

### 4.1 EXPERIMENTAL SETUP

**Environments.** We evaluate our approach across 10 virtual environments that extend from previous EVT benchmark (Zhong et al., 2017; 2024), including FlexibleRoom for training, and rest for testing, as shown in Figure 5. These environments span urban, confined, industrial, and architectural settings with varied challenges. Please refer to the Appendix D for detailed introduction.

**Instruction Set Creation.** To evaluate UC-EVT, we construct an instruction set that includes *sequential instructions* and *target-switch instructions*. Specific details can be found in Appendix F.1. For sequential instructions, we parse the comma-separated sub-instructions and input each subsequent sub-instruction at 120-step intervals during trajectory execution, achieving continuous instruction input throughout the trajectory. For target-switch instructions, since animals perform random walks in the environment, we utilize the environment's absolute information to detect when animals enter the field of view and trigger the corresponding instruction input accordingly.

**Evaluation Metric.** Our ultimate goal is to realise the flexibility and versatility of embodied visual tracking and enable more natural human-robot interactions through our hierarchical aligned agents. Therefore, we keep the consistency with the task definition, assessing our method's ability given diverse textual instructions. In the experiment, we call the UnrealCV API to obtain the tracker's real-time spatial position $(\rho_t, \theta_t)$ and calculate the reward corresponding to different instructions, formulated as: $r(I_t, s_t) = 1 - \frac{|\rho_t - \rho_t^*|}{\rho_{max}} - \frac{|\theta_t - \theta_t^*|}{\theta_{max}}$, where $(\rho_{max}, \theta_{max})$ are the maximum distance and angle within the tracker's field of view, $(\rho_t^*, \theta_t^*)$ are spatial goal corresponding to Instruction $I_t$. We follow the evaluation setting in previous works Zhong et al. (2023; 2024), setting the episode length to 500 steps with corresponding termination conditions. We use three evaluation metrics: 1) Average Accumulated Reward (AR) calculates the average accumulated reward over 50 episodes, indicating overall performance in instruction-behavior alignment; 2) Average Episode Length (EL) is the average number of steps across 50 episodes, reflecting long-term tracking performance. 3) Success Rate (SR) calculates the percentage of episodes reaching 500 steps in 50 episodes.

**Baselines.** To evaluate the tracking performance of our method under instruction settings, we compare it with six representative baselines from classical control, reinforcement learning, and large-model

Table 1: Performance comparison across ten environments using sequential instructions as input. Each cell reports Average Accumulated Reward (AR), Average Episode Length (EL), and Success Rate (SR) in the format AR/EL/SR.

| Environment | Mask-PID (real-time) | | | Ensembled RL (real-time) | | | Word2Vec+RL (real-time) | | | GPT-4o (<1 FPS) | | | TrackVLA (8 FPS) | | | Ours (real-time) | | |
|---|---|---|---|---|---|---|---|---|---|---|---|---|---|---|---|---|---|---|
| | AR↑ | EL↑ | SR↑ | AR↑ | EL↑ | SR↑ | AR↑ | EL↑ | SR↑ | AR↑ | EL↑ | SR↑ | AR↑ | EL↑ | SR↑ | AR↑ | EL↑ | SR↑ |
| FlexibleRoom | 154 | 365 | 0.50 | 183 | 330 | 0.36 | 28 | 357 | 0.42 | 15 | 194 | 0.10 | 57 | 500 | 1.00 | 278 | 500 | 1.00 |
| Suburb | 124 | 386 | 0.36 | 126 | 294 | 0.22 | -5 | 182 | 0.00 | 14 | 241 | 0.20 | -46 | 306 | 0.30 | 166 | 445 | 0.72 |
| Supermarket | 175 | 298 | 0.26 | 114 | 393 | 0.38 | -6 | 186 | 0.00 | 54 | 286 | 0.16 | 35 | 337 | 0.48 | 212 | 422 | 0.64 |
| Parking Lot | 112 | 356 | 0.38 | 169 | 301 | 0.42 | 2 | 192 | 0.00 | 23 | 286 | 0.30 | 70 | 441 | 0.80 | 169 | 491 | 0.93 |
| Old Factory | 80 | 309 | 0.30 | 64 | 334 | 0.36 | -15 | 183 | 0.00 | 5 | 297 | 0.32 | 22 | 390 | 0.60 | 128 | 469 | 0.84 |
| Container Yard | 140 | 369 | 0.42 | 27 | 327 | 0.24 | -28 | 121 | 0.00 | -43 | 149 | 0.00 | -43 | 398 | 0.60 | 156 | 469 | 0.76 |
| Desert Ruins | 128 | 297 | 0.26 | 56 | 368 | 0.34 | 9 | 182 | 0.00 | -5 | 249 | 0.26 | 56 | 421 | 0.72 | 148 | 434 | 0.74 |
| Brass Garden | 120 | 302 | 0.38 | -8 | 348 | 0.34 | -6 | 142 | 0.00 | 3 | 243 | 0.18 | -1 | 324 | 0.44 | 126 | 425 | 0.72 |
| Old Town | 73 | 305 | 0.32 | 14 | 356 | 0.32 | -12 | 196 | 0.00 | 29 | 304 | 0.28 | -133 | 247 | 0.16 | 85 | 433 | 0.64 |
| Roof City | 64 | 289 | 0.34 | 19 | 322 | 0.38 | -8 | 173 | 0.00 | 18 | 256 | 0.20 | -60 | 301 | 0.36 | 86 | 400 | 0.58 |
| **Average (Mean)** | 117.0 | 327.6 | 0.35 | 76.4 | 337. | 0.34 | -4.1 | 191 | 0.04 | 11.3 | 250 | 0.20 | -4.3 | 367 | 0.55 | 155.4 | 448.8 | 0.76 |
| **Average (Std)** | ±34.0 | ±34.9 | ±0.07 | ±64.1 | ±28.5 | ±0.06 | ±14.2 | ±59.6 | ±0.13 | ±23.8 | ±46.1 | ±0.09 | ±61.7 | ±72.7 | ±0.24 | ±54.8 | ±30.6 | ±0.13 |

paradigms: (i) *Mask-PID*: A traditional two-stage tracking paradigm using target masks and a PID controller for spatial goal alignment. (ii) *Ensembled RL Policy*: An extension of the RL-based EVT model (Previous SOTA in ECCV24 Zhong et al. (2024)), where multiple policies are trained for different spatial goals and evaluated using regex-based goal matching. (iii) *Word2Vec+RL*: A variant of Ensembled RL that jointly trains an instruction encoder with a policy network using pretrained Word2Vec embeddings (Mikolov et al., 2013) . (iv) *GPT-4o*: VLM model that generates discrete actions from observed images and natural language instructions through multimodal reasoning. (v) *TrackVLA*: A state-of-the-art VLA-based tracker (Wang et al., 2025) that adapts large vision-language-action models for embodied visual tracking tasks.Details of these baselines are provided in the Appendix G.2.

## 4.2 MAIN RESULTS (Q1)

**Overall Performance** As shown in Table 1, our method achieves the best performance in the FlexibleRoom environment, with a perfect success rate and the highest reward (AR = 278). While TrackVLA also attains a success rate of 1.0, its reward is much lower (AR = 57), reflecting weak alignment despite target retention. Traditional baselines such as Mask-PID , Ensembled RL , and Word2Vec+RL achieve moderate episode lengths but fail to ensure dynamic intention alignment. In contrast, large-model approaches (GPT-4o) suffer from low efficiency and poor responsiveness, leading to significantly degraded tracking.

**Generalization in Unseen Environments** More importantly, our approach maintains robust performance across all nine unseen environments, with success rates ranging from 0.58 (Roof City) to 0.93 (Parking Lot) and episode lengths consistently exceeding 400 steps. This demonstrates strong environmental generalization, in sharp contrast to baselines that degrade severely in unseen settings. Although TrackVLA achieves reasonable tracking in certain environments (e.g., FlexilbeRoom, ParkingLot and Desert Ruins), its performance varies drastically across scenes and reveals weak cross-environment generalization, further underscoring the robustness of HIEVT.

**Baseline Limitations** The comparative analysis reveals distinct failure modes across baselines: (1) Mask-PID benefits from using absolute ground-truth annotations, showing relatively smaller degradation across different environments. However, it still lags far behind HIEVT due to the inherent limitations of regex parsing and PID control. In particular, when the tracking distance or goal changes, the intrinsic oscillatory behavior of PID often causes the target to drift out of view, especially during rapid switches near the frame boundary. (2) Ensemble RL preserves the strengths of RL-based methods (SOTA in ECCV'24), showing good generalization and stability when tracking fixed targets. However, its main limitation lies in handling dynamic goal switches: frequent policy loading incurs inference delays, and temporal feature inconsistency across different policies leads to misalignment between instructions and behavior. (3) GPT-4o, despite strong reasoning capabilities, is constrained by extreme inference latency, resulting in a disastrous performance. (4) TrackVLA, while performing reasonably well in human-centered scenarios, lacks explicit spatial goal representations and shows poor robustness in generalization. It fails to adapt to unseen target categories (Table 3) and quickly degrades under higher target speeds (Table 2), underscoring its limited applicability to UC-EVT;

**Key Advantages** Our approach's superior performance stems from its spatial goal representation serving as an intermediate abstraction between language and action. This design enables consistent

tracking across diverse environments without requiring environment-specific adaptation, while maintaining real-time performance (50 FPS). The results demonstrate that our framework successfully bridges the gap between natural language instructions and precise spatial tracking behaviors in dynamic, real-world scenarios.

## 4.3 ADAPTABILITY OF THE GOAL-ALIGNED POLICY (Q2)

Figure 3: Pixel-level distance between target center and goal center across time steps. The spike at step #101 represents a goal shift instruction, followed by rapid corrections (steps #107, #112) as our agent adjusts to the new spatial goal.

Table 2: **Speed generalization analysis** in FlexibleRoom. Only our method maintains high success rates at high speeds (SR=0.84 at 2.0 m/s) while baselines show severe degradation.

| Method | Maximum moving speed of the target | | |
|---|---|---|---|
| | 0.5 m/s | 1.0 m/s | 2.0 m/s |
| Mask-PID | 154/365/0.50 | -5/330/0.44 | -23/125/0.02 |
| Ensembled RL | 183/330/0.36 | 187/313/0.32 | -183/208/0.10 |
| Word2Vec+RL | 28/357/0.42 | 21/310/0.36 | -56/224/0.12 |
| GPT-4o | 15/194/0.10 | 4/229/0.08 | -119/122/0.00 |
| TrackVLA | 57/**500**/**1.00** | 19/431/0.70 | -2/287/0.30 |
| **Ours** | **278**/**500**/**1.00** | **274**/**496**/**0.98** | **145**/**463**/**0.84** |

**Precise and Fast Alignment.** To demonstrate HIAEVT's precise and efficient adaptation to dynamically changed instructions, we visualised a goal-switch case and recorded the pixel-level deviation in real-world deployment. for quality analysis, as shown in Figure 3. When the goal changes at step #101, our system responds with remarkable speed—reducing the pixel distance from 67 to 24 pixels within just 6 steps (#107) and achieving near-perfect alignment (2 pixels) by step #112. This represents complete adaptation within **220ms**. The system maintains this precision through step #127 despite continued target movement, showcasing both initial responsiveness and sustained tracking accuracy. Table 2 further validates this capability across varying target speeds, with our method maintaining a 0.84 success rate even at 2.0 m/s while all baselines fail (0.00-0.30 success rates).

**Real-time inference challenges (Q3)** We tested system robustness by increasing target speeds from $0.5m/s$ to $2.0m/s$ (Table 2). This experiment reveals a critical real-world challenge: making decisions under real-time constraints. Large-model based methods (TrackVLA, GPT-4o) fail completely at higher speeds (0.30 SR at 2.0m/s) due to inference latency (8 FPS, <1 FPS), while conventional approaches (PID, Ensembled RL) degrade significantly (0.50→0.02 SR, 0.36→0.10 SR). In contrast, our method maintains a high success rate (0.84 SR) even at 2.0m/s. This demonstrates our hierarchical framework successfully balances semantic understanding with operational efficiency—a crucial capability for real-world deployment where targets move at unpredictable speeds and instructions require immediate responses.

Table 3: **Category generalization.** Four useen animals were introduced to FlexibleRoom. Results with dynamic target switching

Table 4: Real-world evaluation of UC-EVT performance on a target person walking in S-patterns for **60 seconds**, starting from three different initial distances (1.5m, 2.0m, 3.0m). We report average IoU between the current and desired bounding box positions and success rate over three trials.

| Method | AR | EL | SR |
|---|---|---|---|
| Mask-based PID | 152 | 364 | 0.52 |
| Ensembled Policy | 76 | 210 | 0.22 |
| Word2Vec+RL | -48 | 92 | 0.00 |
| TrackVLA | 122 | 312 | 0.20 |
| GPT-4o | 18 | 186 | 0.08 |
| **Ours** | **234** | **435** | **0.82** |

| Initial Distance | Avg. IoU | Success Rate |
|---|---|---|
| 1.5m | $0.68 \pm 0.04$ | 0.86 |
| 2.0m | $0.78 \pm 0.05$ | 1.00 |
| 3.0m | $0.72 \pm 0.07$ | 0.98 |

**Adaption to dynamic unseen target switches (Q3).** We further evaluate the ability of baseline methods to handle *dynamic target switching* when unseen target appear in the environment. Specifically, four animals (cow, dog, leopard, horse) were introduced in the FlexibleRoom, and whenever a new animal entered the view, we call the corresponding target-switching instruction (Appendix Table 10). This setting simultaneously tests both instruction comprehension and cross-category generalization.

As shown in Table 3, most baselines experience a sharp performance drop under target switches. Only **HIEVT** and *Mask-PID* maintain relatively stable, with HIEVT achieving the highest success rate (82%). Although *TrackVLA* performs well when tracking humans, its success rate drops drastically on unseen animal categories, reflecting weak cross-category generalization. These results highlight the difficulty of UC-EVT and the effectiveness of HIEVT in handling dynamic, multi-target scenarios.

### 4.4 Ablation Studies (Q4)

We conduct ablation studies (Table 5) to verify each component's contribution. Our findings reveal: 1) **Goal representation**: Vector-based goals maintain reasonable episode lengths but limit precise spatial strategy learning. CLIP-encoded text goals perform poorly due to CLIP's limited semantic-visual aligning capabilities. 2)**Training objectives**: Removing reward regression or IoU-based rewards significantly decreases performance during goal transitions, demonstrating their critical role in maintaining dynamic adaptability. 3) **LLM scaling**: While larger models (GPT-4o vs. Gemma3-27B) generally provide better instruction-goal alignment, even smaller models (Gemma3-1B/4B) achieve strong results, indicating our framework's efficiency across computational constraints. 4) **Memory-Augmented Goal Correction**: We further analyze the impact of memory buffer size, which accumulates history trajectory observations to refine LLM-generated spatial goals. The larger buffer size could store more diverse history trajectories. As shown in Figure 4, larger buffers consistently improve average reward. This demonstrates that Goal Correction effectively adapts high-level goal expectations to the target's actual spatial morphology and dynamics, yielding more accurate and stable tracking. The further sensitive analysis of IoU threshold selection is listed in Appendix F.2.

### 4.5 Results on Real-world Environments (Q5)

To demonstrate the practical applicability and robustness of our goal-behavior alignment framework, we deploy the agent on a mobile wheel robot to handle real-world variability and dynamically adapt to human instructions, the deployment detail and more video clips are available in Appendix J.

**Quantitative Analysis.** Except the quality analysis exhibited in Figure 3. We conducted additional quantitative experiments in real-world settings. We evaluated tracking performance with the target person walking in S-patterns for continuous 60 seconds from three different initial distances (1.5m, 2m, 3m). We measured IoU between the current and initial bounding box positions (target at the desired location) across three trials per distance, shown in Table 4. For each setting, we repeated the experiments five times to ensure robust and reliable performance metrics. These results demonstrate that HIEVT excels in real-world robustness, generalization, and accuracy, successfully adapting to real-world and varying initial conditions while maintaining high tracking performance.

Table 5: Ablation study and model variant performance comparison in FlexibleRoom environment. Each cell shows average reward/episode length/success rate.

Figure 4: The ablation of memory buffer size.

| Method | AR | EL | SR |
|---|---|---|---|
| Ours (GPT-4o) | **274** | **496** | **0.98** |
| Ours (Gemma3-27B) | 230 | 492 | **0.98** |
| Ours (Gemma3-4B) | 179 | 484 | 0.94 |
| Ours (Gemma3-1B) | 144 | 448 | 0.86 |
| w/o spatial goal(Vector) | 27 | 487 | 0.46 |
| w/o spatial goal (CLIP) | 102 | 244 | 0.24 |
| w/o reward regression | 16 | 378 | 0.46 |
| w/o IOU-based reward | -1 | 341 | 0.42 |

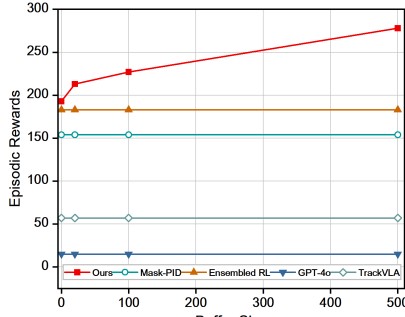

## 5 Conclusion

In this paper, we introduced HIEVT, a hierarchical tracking agent for User-Centric Embodied Visual Tracking that bridges the semantic-spatial gap in human-robot interaction. Experimental results across ten diverse environments demonstrate substantial performance advantages over existing methods, particularly in adaptability and generalization to unseen environments. Our quality analysis and quantitative experiment on real-world robot deployment validate that this hierarchical design effectively balances sophisticated instruction understanding with operational efficiency, providing an elegant and practical solution for user-guided spatial intelligence in embodied systems that scales with minimal additional data requirements.

## ETHICS STATEMENT

The authors have read and acknowledge adherence to the ICLR Code of Ethics. We address potential ethical considerations related to our work below:

**Potential Applications and Societal Impact:** Our instruction-guided visual tracking method is designed for robotic applications in controlled environments such as search and rescue, elderly care assistance, and automated monitoring systems. We acknowledge that tracking technologies could potentially be misused for surveillance purposes that may infringe on privacy rights. However, our work focuses on beneficial applications and we encourage responsible deployment of such technologies with appropriate oversight and consent mechanisms.

**Bias and Fairness:** The virtual environments and synthetic data used in our experiments are designed to be diverse in terms of scenes, lighting conditions, and target entities. However, we acknowledge that the simulated environments may not fully capture the diversity of real-world scenarios and populations. Future work should consider broader environmental and demographic diversity to ensure fair performance across different contexts.

**Environmental Considerations:** Our experiments require computational resources for training and evaluation. We have made efforts to optimize our methods for computational efficiency and provide clear documentation to enable reproducible research without unnecessary resource waste.

**Research Integrity:** All experimental results reported in this paper are obtained through rigorous experimentation with proper statistical analysis. We provide comprehensive implementation details and will make our code available for reproducibility. No conflicts of interest exist among the authors that could bias the research outcomes.

**Data and Code Availability:** We commit to releasing our implementation code and experimental configurations to support reproducible research. The virtual environments used are publicly available through the UnrealZoo platform, ensuring transparency and accessibility for the research community.

The authors believe this work contributes positively to the advancement of embodied AI research while adhering to ethical research practices and acknowledging potential societal implications of the developed technology.

## REPRODUCIBILITY STATEMENT

To ensure the reproducibility of our research, we have made the following efforts and provide comprehensive implementation details:

**Source Code Availability:** We provide complete source code including data collection scripts, training implementations, network architectures and virtual environment download through an anonymous repository `https://anonymous.4open.science/r/Hierarchical-Instruction-aware-Embodied-Visual-Tracking-7357/`. The codebase contains all necessary components to reproduce our experimental results, including hyperparameter configurations and training procedures.

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

# A  APPENDIX

# B  USAGE OF LARGE LANGUAGE MODELS (LLMS)

Large Language Models were used in this work solely as a writing assistance tool. Specifically, we utilized LLMs for: **Language polishing and refinement**: Improving sentence structure, grammar, and overall readability of the manuscript.

**Important clarifications:**

- LLMs were **not** involved in research ideation, methodology design, experimental design, or result interpretation.
- All technical contributions, algorithmic innovations, and scientific insights are entirely the work of the human authors.
- LLMs did not generate any substantial content, figures, tables, or technical descriptions.
- The research direction, experimental methodology, and all conclusions were independently developed by the authors.
- LLM assistance was limited to linguistic improvements of already-written content, with all technical and scientific content remaining unchanged.

The role of LLMs in this work was purely editorial and did not contribute to the research contributions or scientific merit of the paper.

# C  PRELIMINARIES

**Problem Definition.**  User-Centric Embodied Visual Tracking (UC-EVT) starts with an initial user instruction and a target. The tracker then attempts to track the target immediately by following the user's instruction.The main components of this problem are outlined as follows:

- **User Instruction.** The user initially provides an instruction $\mathcal{I}_0$, which serves as the starting command for the tracking agent. At any subsequent time step $t$, the user can issue a new instruction $\mathcal{I}_t$ to update the agent's behavior.
- **Tracking State.** The tracking agent's state at time $t$, denoted as $s_t$, is represented by the relative distance $\rho_t$ and the relative angle $\theta_t$ with respect to the target. Specifically, the tracking state is given as: $\mathcal{S}_t = (\rho_t, \theta_t)$.
- **Agent Action.** At each time step $t$, the tracking agent executes an action $a_t$ to adjust its position and orientation in order to minimize the discrepancy between the current tracking state and the user's instruction.
- **Environment Transferring.** To ensure applicability to real-world scenarios, the tracker should be tested in unseen environments. This setup is designed to test the generalization and transferability of the tracking system.

**Objective Function.**  The goal of UC-EVT is to minimize the discrepancy between the user instruction $\mathcal{I}_t$ and the tracking state $s_t$ throughout the entire tracking period $T$. This discrepancy is measured using a distance function $\mathcal{D}(\mathcal{I}_t, s_t)$. The objective function is defined as $\min \sum_{t=1}^{T} \mathcal{D}(\mathcal{I}_t, s_t)$, where $\mathcal{D}(\mathcal{I}_t, s_t)$ quantifies the difference between the user instruction $\mathcal{I}_t$ and the tracking state $s_t$ at each time step $t$. The objective aims to ensure that the tracking agent follows the user's instructions as closely as possible over the entire tracking period (episode).

To better difference UC-EVT task with previous EVT task, we present a clearer comparison, shown in Table 6.

Table 6: Comparison between Traditional EVT and User-Centric EVT

| Aspect | Traditional EVT | UC-EVT (Ours) |
|---|---|---|
| User Control | System restart for new target | Natural language during tracking |
| Tracking Distance | Static | Changeable |
| Viewing Angle | Static | Changeable |
| Interaction Frequency | Only at initialization | Continuous |

# D  VIRTUAL ENVIRONMENT

Our experiments were conducted across 10 diverse virtual environments built using Unreal Engine and integrating UnrealCV (Qiu et al., 2017) for programmatic control. Specifically, we extend from the previous public EVT benchmark Gym-UnrealCV (Zhong et al., 2024), adopting the original FlexibleRomm (SimpleRoom in EVT benchmark) for training, ParkingLot as an unseen environment for testing. The remaining 8 unseen environments serve as an enlargement of the original EVT benchmark environments, which increase the challenge in terms of visual dynamic realism and context complexity. The enlarged environments are developed based on UnrealZoo (Zhong et al., 2025), aiming to evaluate different aspects of instruction-aware tracking under various challenging conditions. Environment binary could be downloaded from `https://modelscope.cn/datasets/UnrealZoo/UnrealZoo-UE4`, and code are available in `https://anonymous.4open.science/r/Hierarchical-Instruction-aware-Embodied-Visual-Tracking-7357/`. Though some recent simulators provide human following functionalities, such as Habitat (Puig et al., 2023), we find their environmental diversity and task configuration are limited, incapable of comprehensively evaluating the UC-EVT task, here we compare them in Table 7, demonstrating the necessity of the benchmark extension.

Table 7: Comparison of UC-EVT with Related Benchmarks

| Aspect | UC-EVT | Habitat 3.0 | Gym-UnrealCV |
|---|---|---|---|
| Environment | indoor+outdoor | indoor | indoor+outdoor |
| Humanoid | ✓ | ✓ | ✓ |
| Unseen Targets | ✓ | ✗ | ✓ |
| Speed Variant | ✓ | ✗ | ✗ |
| Distance config. | ✓ | ✗ | ✗ |
| Angle config. | ✓ | ✗ | ✗ |
| Target select. | ✓ Anytime | ✗ | ✓ System init |
| Goal Spec. | ✓ Natural language | ✗ | ✗ |
| Dynamic switch goal | ✓ | ✗ | ✗ |

## D.1  TRAINING ENVIRONMENT

FlexibleRoom: This environment, adopted from previous work Zhong et al. (2023; 2021), serves as our primary training venue. It features an adaptable indoor space with programmable lighting conditions, furniture layouts, and target navigation patterns. The environment's built-in navigation system enables automatic generation of diverse trajectories through randomly sampled destinations, which is particularly valuable for data collection in goal-conditioned reinforcement learning. We extended this environment with customizable appearance factors like texture, lighting, and furniture placement to enhance training diversity and reduce overfitting. The modular design allows us to systematically control visual complexity while maintaining consistency in the underlying spatial relationships.

## D.2  TESTING ENVIRONMENT

**Suburb:** A meticulously designed suburban neighborhood featuring irregular terrain, diverse vegetation, and dynamic obstacles that simulate pedestrian and vehicular movement. This environment tests the agent's ability to maintain tracking across changing elevation, lighting conditions, and partial occlusions from trees and structures. The open spaces combined with clustered obstacles create complex tracking scenarios with variable target visibility.

**Supermarket:** An indoor retail environment with intricate item shelves, static displays, and narrow aisles that closely mimic real-world shopping scenarios. The dense arrangement of objects creates numerous occlusion challenges and confined spaces for navigation. This environment evaluates the system's performance in crowded indoor settings where the target frequently disappears behind shelves and reappears elsewhere.

**Parking Lot:** An outdoor environment featuring multiple parked vehicles under dim lighting conditions. The uniform structure combined with low visibility areas and complex shadows tests the agent's ability to discriminate targets in visually challenging scenes. The environment transitions between open areas and confined spaces between vehicles, requiring adaptive tracking strategies.

**Old Factory:** A deteriorated industrial setting characterized by numerous steel pillars, scattered wooden crates, and uneven lighting. The environment features high ceilings with exposed structural elements and complex shadows that create challenging visual conditions. The combination of open factory floor areas and cluttered storage zones tests the agent's ability to track across rapidly changing visual contexts.

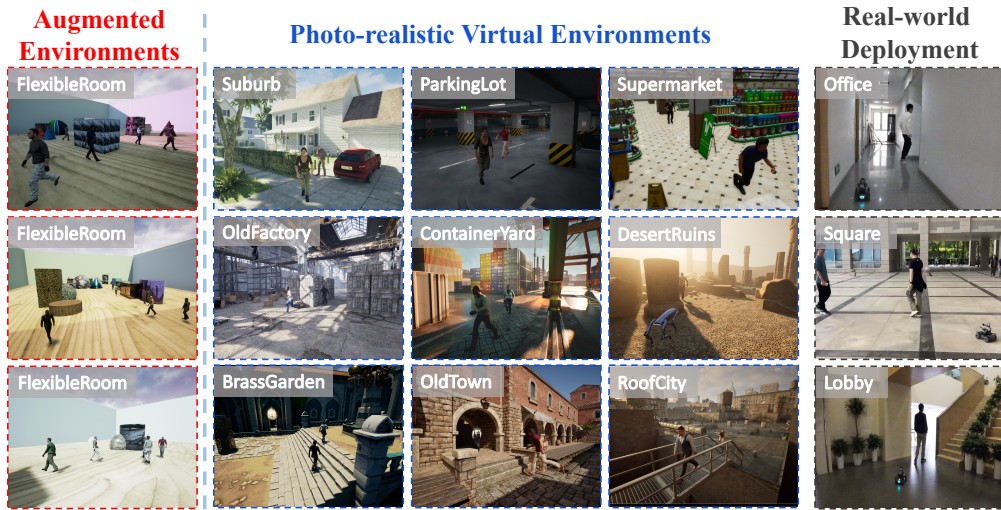

Figure 5: The examples of virtual and real-world environments used in our experiments. The FlexibleRoom environment is used for training data collection, featuring diverse augmentable factor. the nine photo-realistic environments in the middle are used for quantitative evaluation, we also deploy our proposed method on three real-world scenarios to validate the effectiveness and transferability.

**Container Yard:** A dynamic logistics environment featuring stacked shipping containers under changing lighting conditions. The geometric regularity of the containers combined with dramatic lighting variations creates challenging perception scenarios. The environment features narrow corridors between container stacks that frequently occlude targets, testing the system's ability to predict movement through temporary visual obstruction.

**Desert Ruins:** An archaeological site set in harsh desert lighting conditions with scattered walls and pillars creating a complex spatial layout. The environment combines open areas with confined passages and features extreme lighting contrasts between shadow and direct sunlight. This tests the agent's robustness to challenging lighting conditions and irregular spatial structures.

**Brass Gardens:** A palace-style architectural complex featuring narrow corridors and multi-level platforms connected by staircases. This environment uniquely tests non-planar tracking capabilities, as targets frequently change elevation while moving through the environment. The ornate architectural elements and varying ceiling heights create complex spatial reasoning challenges for maintaining consistent tracking.

**Old Town:** A European-style hillside village with interconnected indoor and outdoor spaces linked by narrow, undulating stairways. This environment combines both open plazas and confined interior spaces, requiring frequent adaptation to changing spatial contexts. The irregular layout with multiple elevation changes tests the agent's ability to maintain tracking continuity across diverse architectural spaces.

**Roof City:** A rooftop cityscape featuring protruding air ducts, walkways, and scattered debris that create a maze-like environment. The constrained navigation paths combined with varying elevation levels test the agent's ability to predict movement in spatially restricted areas. The urban setting also introduces complex background textures and challenging lighting conditions from reflective surfaces.

# E  LOW-LEVEL POLICY

## E.1  GOAL RANDOMIZATION AND DATA COLLECTION

In our experiment setting, the tracking goal's relative spatial position is constrained within the range of $\rho^* \in (200, 600)$ and $\theta^* \in (-25°, 25°)$. We uniformly sample tracking goals from this range, resulting in an offline dataset of **1,750,000** steps used to train our proposed method. In this section, we introduce the details of the data collection process, including player initialization, and state-based PID controller with noise perturbation for tracker and trajectory generation.

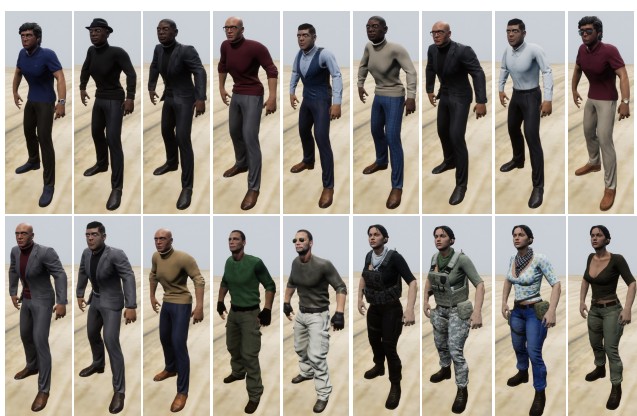

Figure 6: 18 humanoid models are used in data collection and evaluation.

### E.2 PLAYER INITIALIZATION

We use 18 humanoid models as targets and trackers, as shown in Figure 6. At the beginning of each episode, we randomly sample their appearance from the 18 humanoid models, and the target player will be randomly placed in the tracker's visible region.

### E.3 STATE-BASED PID CONTROLLER WITH MULTI-LEVEL PERTURBATION

We first use PID controllers to enable the agent to follow a target object, maintaining a specific distance and relative angle(e.g., 3 meters directly in front of the agent). The process is as follows:

- Setpoints: Define the desired distance and angle as the setpoints for the PID controller (e.g., 3 meters for distance and 0 degrees for angle).

- Process Variables: Measure the actual distance and angle between the agent and the target object using the grounded state data accessible via the UnrealCV API. These measurements serve as the process variables for the PID controller.

- Error Calculation: Calculate the error between the setpoints and the process variables, which will be used as inputs for the PID controller.

- Control Output: Apply the PID equation to generate the control output, determining the agent's speed and direction:

$$u(t) = K_p e(t) + K_i \int_0^t e(\tau)d\tau + K_d \frac{de(t)}{dt} \tag{6}$$

where $u(t)$ is the control output, $e(t)$ is the error, $K_p$, $K_i$, and $K_d$ are the proportional, integral, and derivative gains, respectively.

- Fine-tune the PID gains to achieve optimal controller performance. For instance, increasing $K_p$ will enhance the agent's responsiveness to errors but may introduce overshoot or oscillations. Raising $K_i$ helps minimize steady-state error but can lead to integral windup or slower response. Boosting $K_d$ reduces overshoot and dampens oscillations but may also amplify noise or cause derivative kick. If the control output exceeds the defined action space limits, it is clipped. The tuned gains are detailed in Table 8.

Table 8: The parameters we used in the state-based PID controller.

| Controller | $K_p$ | $K_i$ | $K_d$ |
|---|---|---|---|
| Speed | 5 | 0.1 | 0.05 |
| Angle | 1 | 0.01 | 0 |

Then, we introduce noise perturbation to the PID output, causing the agent to alternately deviate from and recover towards the desired distance and angle. This set-up aims to collect trajectories with diverse step rewards, alleviating the overestimation problem during offline training. We set a threshold $p = 0.15$, and if the probability value at time $t$ is greater than $p$, which is $P(t) > p$, the agent takes a random action from the action space and

continues for $L$ steps. We also adopt a random strategy to set the step length $L$, with an upper limit of 4. After the random actions of the agent end, we use a random function to determine the duration of the next random action $L$. Here, we set the upper limit to 4 because we found that the number of times the agent failed in a round significantly increased beyond 4. Therefore, we empirically set the upper limit of the random step length to 4.

# F    HIGH-LEVEL SEMANTIC-SPATIAL GOAL ALIGNER

During training phase, we implement goal randomization within a continuous space, within the range of $\rho^* \in (200, 600)$ and $\theta^* \in (-25°, 25°)$. However, due to computational constraints, it is impractical to evaluate all points within this continuous space. Therefore, we constructed an instruction list through a two-step process: first, we defined a sequence of spatial position transitions; then, we generated corresponding language instructions for each transition.

## F.1    INSTRUCTIONS GENERATION

To fairly and accurately evaluate our method's ability to align with human-like real world instructions, we generated a list of sequential text instructions by first defining various spatial objective transitions (right column in Table 10), then we generate a corresponding natural language instruction by leveraging GPT-4o API (left column in Table 10). These instructions, as shown in Table 10, are designed to reflect both absolute and relative position changes, as well as some ambiguous representation and target-switching behaviors. The instructions are organized into two main categories: sequential instructions and target-switching instructions:

**Sequential instructions** are designed to evaluate the model's ability to dynamically understand and align with a sequence of instructions over time. These instructions require the agent to follow a series of spatial objectives, each dependent on the previous one. By evaluating the agent's performance with sequential commands, we test the system's capacity to adapt its behavior as the instruction set evolves, ensuring that the model can consistently track and follow a set of changing goals.

**Target-switching instructions**, on the other hand, are aimed at evaluating the model's dynamic generalization ability to different target categories under changing conditions. These instructions ask the agent to switch focus from one target (e.g., "the person") to another (e.g., "the beagle"), while maintaining consistent tracking and navigation behavior.

**To balance natural language variation with consistent evaluation metrics**, we manually map these instructions to the expected absolute or relative spatial transitions for each instruction. This predefined mapping serves as a benchmark, allowing for objective comparison between the agent's actions and the desired user intent.

## F.2    SENSITIVITY EXPERIMENT ON IOU THRESHOLD SELECTION

The threshold of 0.5 in the equation 3 is empirically set. The IoU threshold aims to correct unreasonable goals while tolerating tiny deviations. We conduct additional sensitivity experiments, shown in Table 9. Performance remains robust across IoU thresholds [0.4, 0.7], with optimal results at 0.5 (AR=278, SR=1.0). Thresholds below 0.4 compromise tracking precision, while values above 0.7 overly penalize minor deviations. We adopt 0.5 to balance accuracy and robustness.

Table 9: IoU Threshold Sensitivity Analysis

| IoU Threshold | AR | EL | SR |
|---|---|---|---|
| 0.3 | 246 | 481 | 0.92 |
| 0.4 | 259 | 494 | 0.98 |
| 0.5 | 278 | 500 | 1.0 |
| 0.6 | 270 | 500 | 1.0 |
| 0.7 | 271 | 489 | 0.96 |

Table 10: Examples of instruction pools, including sequential instructions and target switch instructions, along with the corresponding spatial goal transitions.

| Instruction | Corresponding Spatial Goal Transition |
|---|---|
| *Sequential Instructions* | |
| Keep the person in the close center, move slightly to the left, increase the distance, keep the person at the far center. | target $\leftarrow$ person, $(200, 0°)$ $\rightarrow$ $(200, -20°)$ $\rightarrow$ $(350, -20°) \rightarrow (450, 0°)$ |
| Maintain the person on the right, bring the person closer, stay roughly near the center, keep the person aligned to the left. | target $\leftarrow$ person, $(350, 20°)$ $\rightarrow$ $(200, 20°)$ $\rightarrow$ $(350, 0°) \rightarrow (350, -20°)$ |
| Keep the person in the far-away center, shift a bit to the right, make sure it is not too far away, keep the person in the close center. | target $\leftarrow$ person, $(450, 0°) \rightarrow (450, 20°) \rightarrow (\rho^* \leq 350, 20°) \rightarrow (200, 0°)$ |
| Position the person on the left, move the person closer, stay closer but still on the left side, keep the person in the center at close range. | target $\leftarrow$ person, $(350, -20°)$ $\rightarrow$ $(200, -20°)$ $\rightarrow$ $(\rho^* \leq 250, -20°) \rightarrow (200, 0°)$ |
| Keep the person directly ahead, move slightly to the right, increase the distance, keep the person far on the right. | target $\leftarrow$ person, $(350, 0°)$ $\rightarrow$ $(350, 20°)$ $\rightarrow$ $(450, 20°) \rightarrow (450, 20°)$ |
| Ensure the person is on the left, reduce the distance, stay roughly near the center, maintain the person at medium range in front. | target $\leftarrow$ person, $(350, -20°)$ $\rightarrow$ $(200, -20°)$ $\rightarrow$ $(350, 0°) \rightarrow (350, 0°)$ |
| Keep the person near the center, step back slightly, move slightly to the left, keep the person far on the left side. | target $\leftarrow$ person, $(200, 0°)$ $\rightarrow$ $(350, 0°)$ $\rightarrow$ $(350, -20°) \rightarrow (450, -20°)$ |
| Keep the person in the close right, increase the distance, shift slightly to the left, keep the person at the far center. | target $\leftarrow$ person, $(200, 20°)$ $\rightarrow$ $(350, 20°)$ $\rightarrow$ $(350, 0°) \rightarrow (450, 0°)$ |
| Position the person at medium distance in the center, move slightly to the right, reduce the distance, keep the person close in the center. | target $\leftarrow$ person, $(350, 0°)$ $\rightarrow$ $(350, 20°)$ $\rightarrow$ $(200, 20°) \rightarrow (200, 0°)$ |
| Keep the person in the far left, move closer, stay roughly in front, keep the person at medium range in the center. | target $\leftarrow$ person, $(450, -20°)$ $\rightarrow$ $(350, -20°)$ $\rightarrow$ $(350, 0°) \rightarrow (350, 0°)$ |
| Start with the person at close right, gradually increase distance, move to center alignment, then position far to the left. | target $\leftarrow$ person, $(200, 20°)$ $\rightarrow$ $(350, 20°)$ $\rightarrow$ $(400, 0°) \rightarrow (450, -20°)$ |
| Position the person at medium right, increase distance while maintaining angle, shift to left alignment, bring closer to medium left. | target $\leftarrow$ person, $(350, 20°)$ $\rightarrow$ $(450, 20°)$ $\rightarrow$ $(450, -20°) \rightarrow (350, -20°)$ |
| Keep the person close center, move to close right, step back to medium distance, finally position far right. | target $\leftarrow$ person, $(200, 0°)$ $\rightarrow$ $(200, 20°)$ $\rightarrow$ $(350, 20°) \rightarrow (450, 20°)$ |
| Maintain person at close left, step back gradually, center the alignment, continue stepping back to far center. | target $\leftarrow$ person, $(200, -20°)$ $\rightarrow$ $(350, -20°)$ $\rightarrow$ $(350, 0°) \rightarrow (450, 0°)$ |
| Position person at medium center, shift to medium left, increase distance to far left, then center while maintaining far distance. | target $\leftarrow$ person, $(350, 0°)$ $\rightarrow$ $(350, -20°)$ $\rightarrow$ $(450, -20°) \rightarrow (450, 0°)$ |
| Keep person far right, bring significantly closer to close right, shift toward center, maintain close center position. | target $\leftarrow$ person, $(450, 20°)$ $\rightarrow$ $(200, 20°)$ $\rightarrow$ $(200, 10°) \rightarrow (200, 0°)$ |
| Start with person close right, move to medium right, shift to medium center, then extend to far center. | target $\leftarrow$ person, $(200, 20°)$ $\rightarrow$ $(350, 20°)$ $\rightarrow$ $(350, 0°) \rightarrow (450, 0°)$ |
| Position person at far left, bring to medium left, center the alignment, then move closer to close center. | target $\leftarrow$ person, $(450, -20°)$ $\rightarrow$ $(350, -20°)$ $\rightarrow$ $(350, 0°) \rightarrow (200, 0°)$ |
| Keep person at medium left, extend to far left, shift toward center while maintaining distance, bring closer to medium center. | target $\leftarrow$ person, $(350, -20°)$ $\rightarrow$ $(450, -20°)$ $\rightarrow$ $(450, 0°) \rightarrow (350, 0°)$ |

**Table 10 – continued from previous page**

| Instruction | Corresponding Spatial Goal Transition |
|---|---|
| Start close center, move to close left, extend distance to medium left, further extend to far left. | target $\leftarrow$ person, $(200, 0^\circ)$ $\rightarrow$ $(200, -20^\circ)$ $\rightarrow$ $(350, -20^\circ) \rightarrow (450, -20^\circ)$ |
| Position person far right, shift slightly toward center, bring much closer, maintain close center-right. | target $\leftarrow$ person, $(450, 20^\circ)$ $\rightarrow$ $(450, 10^\circ)$ $\rightarrow$ $(250, 10^\circ) \rightarrow (200, 15^\circ)$ |
| Keep person close left, extend to medium left, shift to medium right, bring closer to close right. | target $\leftarrow$ person, $(200, -20^\circ)$ $\rightarrow$ $(350, -20^\circ)$ $\rightarrow$ $(350, 20^\circ) \rightarrow (200, 20^\circ)$ |
| Start medium center, shift to medium right, extend to far right, center while maintaining far distance. | target $\leftarrow$ person, $(350, 0^\circ)$ $\rightarrow$ $(350, 20^\circ)$ $\rightarrow$ $(450, 20^\circ) \rightarrow (450, 0^\circ)$ |
| *Target-Switching Instructions* | |
| "Switch to tracking the beagle, keep it on the right." | target $\leftarrow$ dog, $(350, 20^\circ)$ |
| "Track the dog with a far distance." | target $\leftarrow$ dog, $(450, 0^\circ)$ |
| "Follow the beagle from the left side." | target $\leftarrow$ dog, $(350, 20^\circ)$ |
| "Follow the cow now, keep it on the left." | target $\leftarrow$ cow, $(350, -20^\circ)$ |
| "Track the cow at a safe distance. " | target $\leftarrow$ cow, $(450, 0^\circ)$ |
| "Keep your eye on the cow " | target $\leftarrow$ cow, $(350, 0^\circ)$ |
| "Tracking the leopard, keep it on a far distance." | target $\leftarrow$ leopard, $(450, 0^\circ)$ |
| "Follow the leopard from the left side, keep a safe distance." | target $\leftarrow$ leopard, $(450, 20^\circ)$ |
| "Stop following the person, track the horse." | target $\leftarrow$ horse, $(350, 0^\circ)$ |
| "Look at the horse, follow it." | target $\leftarrow$ horse, $(350, 0^\circ)$ |
| "keep following the horse from right side." | target $\leftarrow$ horse, $(350, -20^\circ)$ |
| "Switch back to the person at close range." | target $\leftarrow$ person, $(200, 0^\circ)$ |

## G  IMPLEMENTATION DETAILS

In this section, we detail the implementation of our basic setup, baseline methods, and proposed policy network structures.

Table 11: The fine-tuned parameter for Bbox-based PID Controller.

| Controller | $K_p$ | $K_i$ | $K_d$ |
|---|---|---|---|
| Speed | 0.2 | 0.01 | 0.03 |
| Angle | 0.05 | 0.01 | 0.1 |

### G.1  BASIC SETTING

In our experiments, we utilize a continuous action space for agent movement control. The action space comprises two variables: the angular velocity, ranging from $(-30^\circ/s, 30^\circ/s)$ and the linear velocity, ranging from $(-1\,m/s$ ,$1\,m/s)$. We train our models using the Adam optimizer with a learning rate of 3e-5 and a batch size of 128.

### G.2  BASELINE METHODS

Mask-PID: A traditional two-stage tracking paradigm. A PID controller aligns the mask with the instructed spatial goal, while a simple regex-based parser maps natural language directives to predefined goals. To facilitate the influence of different vision models Lei et al. (2025); Liu et al. (2023); Cheng et al. (2023b), focusing on the limitation of traditional tracking paradigm, we leverage the virtual environment built on UnrealEngine and the unrealcv api Qiu et al. (2017) to generate the ground truth target's mask, using predefined bounding box images as the goal representation, adjusting the agent's actions to maximize the intersection-over-union (IoU) with the target mask. We implemented two PID controllers to jointly control the agent's movement: one for linear velocity and one for angular velocity. For linear velocity($V_{linear}$), we use the areas of the target bounding box mask $A_s$ detected from state $s$, and goal bounding box masks $A_{goal}$ as input variables, the $V_{linear}$ is calculated as:

$$V_{linear}(t) = K_p(A_{goal} - A_s(t)) + K_i \int_0^t (A_{goal} - A_s(\tau))d\tau$$
$$+ K_d \frac{d(A_{goal} - A_s(t))}{dt} \tag{7}$$

where $K_p, K_i$, and $K_d$ are the proportional, integral, and derivative gains, respectively. The linear velocity is constrained within the range $(-100, 100)$ to match the training setup. A positive $V_{linear}$ moves the agent forward if $A_s$ is smaller than $A_{goal}$.

For angular velocity $(V_{ang})$, the PID controller aims to minimize the angular deviation by computing the difference between the x-axis coordinates of the centers of the target mask $x_s$ and the goal mask $x_c$, the visualization is shown in Figure 7. The control input $V_{ang}$ is given by:

$$V_{ang}(t) = K_p(x_c - x_s(t)) + K_i \int_0^t (x_c - x_s(\tau))d\tau$$
$$+ K_d \frac{d(x_c - x_s(t))}{dt}$$

(8)

where $K_p, K_i$, and $K_d$ are the proportional, integral, and derivative gains, respectively. The angular velocity is constrained within the range $(-30, 30)$. A positive $V_{ang}$ results in a rightward rotation if $x_s$ is to the right of $x_c$. The final tuned parameters are detailed in the Table 11.

**Ensembled RL:** An extension of the state-of-the-art RL-based EVT model(ECCV24), where multiple policies are trained under different spatial goals $(\rho^*, \theta^*)$, and a regular expression (regex) is used during evaluation to select the policy that best matches the goal inferred from natural language instructions. Specifically, We trained four separate policy networks corresponding to four representative discrete goals:$[g_{close}, g_{far}, g_{left}, g_{right}]$. The corresponding data for these goals were extracted and filtered from the 1750000 steps offline dataset, providing a focused dataset for training ensemble policies. The policy network architecture is based on the latest state-of-the-art method for Embodied Visual Tracking (EVT) as described by Zhong et al. (2024). During evaluation, we choose the corresponding policy based on the goal spatial position indicated by the instruction.

**Word2Vec+RL:** A variant of the Ensembled RL baseline where we jointly train an instruction encoder by applying pretrained word2vec-google-news-300 Mikolov et al. (2013) with a policy network, instead of relying on regex mapping. In implementation, we use the same dataset to train an end-to-end model using word2vec-google-news-300 Mikolov et al. (2013) for text encoding (W2V), a basic four-layer convolutional neural network for image encoding (CNN), with three consecutive observation frames concatenated as input, and a two-layer MLP for feature alignment and action prediction. We use the same Conservative-Q learning method for offline RL training. This approach highlights the limitations of directly combining existing NLP and neural network modules in an end-to-end fashion.

**GPT4-o:** We leverage the multi-modal capabilities of GPT4-o to directly generate actions based on the observed image and the desired goal. To ensure smooth and accurate transitions, we developed a system prompt that aids the large model in comprehending the task and regularizes the output format, aligning it with our predefined action settings. This prompt serves as a guiding framework, enabling the model to produce actions that are coherent with the requirements of the task. Specifically, we converted the bounding box goal representation into text-based coordinates, and the input image was first transformed to the same text-conditioned segmentation mask in the paper, then detected the target's bounding box coordinates as input of LLM. Note that the system prompt could directly use image observation as input, but from our experience, the alignment performance is quite poor. The system prompt content is shown by Figure 12.

**TrackVLA:** A recent the state-of-the-art VLA-based tracker Wang et al. (2025) that adapts large vision-language-action models to embodied visual tracking tasks. In implementation, we obtained the pre-trained model of TrackVLA through direct communication with the original authors via email Wang et al. (2025). This allowed us to directly evaluate the model in our environment using the pre-trained weights without requiring additional training. The only modification we made to the original TrackVLA implementation was adjusting the maximum speed during varying speed testing. Specifically, when testing at 2 m/s, we increased TrackVLA's speed limit to 220 to meet the dynamic requirements (slightly higher than the target's speed for redundancy considerations).

## G.3 POLICY NETWORK

In our approach, we use convolutional neural networks (CNN) as the visual extraction module, which is followed by a fully connected layer and a Reward Head. The output of the reward head is used for reward regression during training. The output of the fully connected layer is fed into the recurrent policy network. We use a long-short-term memory(LSTM) network to model temporal consistency. The recurrent policy network is an extension of the CQL-SAC algorithm Kumar et al. (2020), where we have modified the data sampling and optimization processes to accommodate the LSTM network. The visual temporal features extracted by the CNN and LSTM are then passed to the actor and critic networks, each consisting of two fully connected layers. The hyperparameters and neural network structures employed in our method are detailed in Table 12 and Table 14.

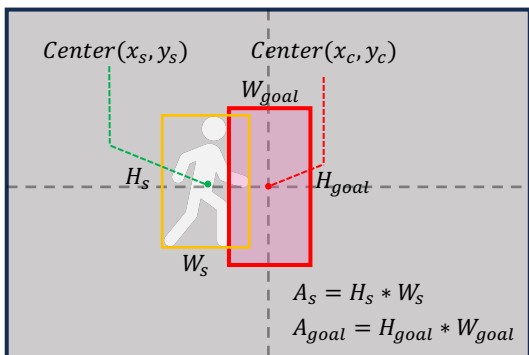

Figure 7: Illustration of the target bounding box (yellow box) and goal bounding box (red box) used in PID controller. The center of target bounding box $Center(x_s, y_s)$ and spatial goal $Center(x_c, y_c)$ are used to calculate horizontal deviations. The area size of of target bounding box $A_s$ and spatial goal $A_{goal}$ are used to calculate distance deviations.

Table 12: The hyper-parameters used for offline training and the policy network.

| Name | Symbol | Value |
|---|---|---|
| Learning Rate | $\alpha$ | 3e-5 |
| Discount Factor | $\gamma$ | 0.99 |
| Batch Size | - | 128 |
| LSTM update step | - | 20 |
| LSTM Input Dimension | - | 256 |
| LSTM Output Dimension | - | 64 |
| LSTM Hidden Layer size | - | 1 |
| Reward Head Input Dimension | - | 256 |
| Reward Head Ouput Dimension | - | 1 |

# H  LLM-BASED SEMANTIC-SPATIAL GOAL ALIGNER

In our method, we develop a hierarchical instruction parser, integrated with Large Language Models (LLM) and chain-of-thought to translate human instructions into a mid-level goal representation. We first describe our evaluation method, then visualize the evaluation result via a radar chart.

## H.1  QUALITY EVALUATION

To evaluate the effectiveness of our instruction parser and investigate the impact of different reasoning mechanisms on the accuracy of bounding box generation, we employ three different methods: GPT-4o: We use GPT-4o with a system prompt that includes both a task introduction and chain-of-thought (CoT) reasoning guidelines for evaluation. GPT-4o w/o CoT: We use GPT-4o with a system prompt containing only the task introduction for evaluation. GPT-o1: We use GPT-o1 with a system prompt that includes the task introduction for evaluation.

We ensure a fair evaluation by sampling 140 instructions from the instruction list in Table **??**, and each instruction is paired with a corresponding spatial goal. This setup minimizes ambiguity in the textual instructions, providing a baseline for evaluating the correctness of the generated bounding boxes. The performance is assessed by the accuracy of generated bounding boxes. The accuracy is calculated as: the number of correctly generated bounding boxes divided by the total sampled instances

For each instruction, we define two categories based on the type of spatial instruction: absolute spatial positions and relative spatial position changes. The rules for evaluating bounding box correctness are as follows:

1)Instructions conveying absolute spatial positions (first 22 rows of Table 4): We use the final generated bounding box $[x, y, w, h]$, calculate horizontal position $x$ and area size $w * h$ to determine correctness.

- $(\rho^*, \theta^*) = (200, 0°)$: Valid if area size is within $(0.06, 0.3)$ and $x$ is within $(0.4, 0.6)$.
- $(\rho^*, \theta^*) = (450, 0°)$: Valid if area size is within $(0, 0.06)$ and $x$ is within $(0.4, 0.6)$
- $(\rho^*, \theta^*) = (350, -20°)$: Valid if $x$ is within $(0, 0.4)$.

Table 13: We directly map our adopted action space (continuous actions) from virtual to real. The second and the third columns are the value ranges of velocities in the virtual and the real robot, respectively.

| Bound of Action | Linear | |
| --- | --- | --- |
| | Virtual (cm/s) | Real (m/s) |
| High | 100, 30 | 0.5, 1.0 |
| Low | -100, -30 | -0.5, -1.0 |
| | Angular | |
| | Virtual (degree/s) | Real (rad/s) |
| High | 100, 30 | 0.5, 1.0 |
| Low | -100, -30 | -0.5, -1.0 |

Table 14: The neural network structure, where $8\times8$-16S4 means 16 filters of size $8\times8$ and stride 4, FC256 indicates fully connected layer with dimension 256, Reward Head 1 means the fully connected layer for reward regression layer with output dimension 1, and LSTM64 indicates that all the sizes in the LSTM unit are 64.

| Module | Goal-state Aligner | | | | Recurrent Policy | | |
| --- | --- | --- | --- | --- | --- | --- | --- |
| Layer# | CNN | CNN | FC | Reward Head | LSTM | FC | FC |
| Parameters | $8\times8$-16$S4$ | $4\times4$-32$S2$ | 256 | 1 | 64 | 2 | 2 |

- $(\rho^*, \theta^*) = (350, 20°)$: Valid if $x$ is within $(0.6, 1)$.

2)Instructions conveying relative spatial position change (last 20 rows of Table 4): We evaluated based on bounding box increments $[\Delta x, \Delta y, \Delta w, \Delta h]$ generated by the parser.

- $(\Delta\rho, \Delta\theta) = (-150, 0°)$: Valid if both $\Delta w$ and $\Delta h$ are positive.
- $(\Delta\rho, \Delta\theta) = (150, 0°)$: Valid if both $\Delta w$ and $\Delta h$ are negative.
- $(\Delta\rho, \Delta\theta) = (0, -20°)$: Valid if $\Delta x < 0$.
- $(\Delta\rho, \Delta\theta) = (0, 20°)$: Valid if $\Delta x > 0$.

The results in Figure 8 confirm that the chain-of-thought (CoT) reasoning mechanism significantly improves the accuracy of the bounding box generation. Compared with GPT-4o w/o CoT and GPT-o1, GPT-4o consistently exceeds 80%, achieving up to 100% accuracy in some cases, demonstrating its reliability in translating textual instructions into spatially accurate bounding boxes. We argue that CoT enhances the model's ability to handle more nuanced spatial relationships. GPT-O1 performs significantly worse than GPT-4o and GPT-4o w/o CoT, and we believe this is largely due to the underlying reasoning mechanism. GPT-o1 employs an automatic decomposition reasoning approach, which decomposes tasks into smaller steps without considering the broader context. This lack of holistic reasoning leads to poor performance in our task, particularly when handling complex spatial relationships.

## H.2 PROMPTS

We define a system prompt aiming to help the LLM understand the tracking task and introduce the Chain-of-thought (CoT) to enhance the LLM's understanding ability. The detailed content of the system prompt is shown in Figure 11.

## I GRAPHIC USER INTERFACE

To enable an intuitive visual interaction and multi-modal instruction input, we design a simple GUI for user input instructions while observing the environment from the tracking agent's first-person view. Users could directly type text instructions and click the "send" button to update instructions. The GUI demonstration is shown in Figure 10.

## J REAL-WORLD DEPLOYMENT

We transfer our agent into real-world scenarios to verify the practical contribution and the effectiveness of our proposed method. Specifically, we use **SAM-Track** Cheng et al. (2023b) to generate segmentation masks from

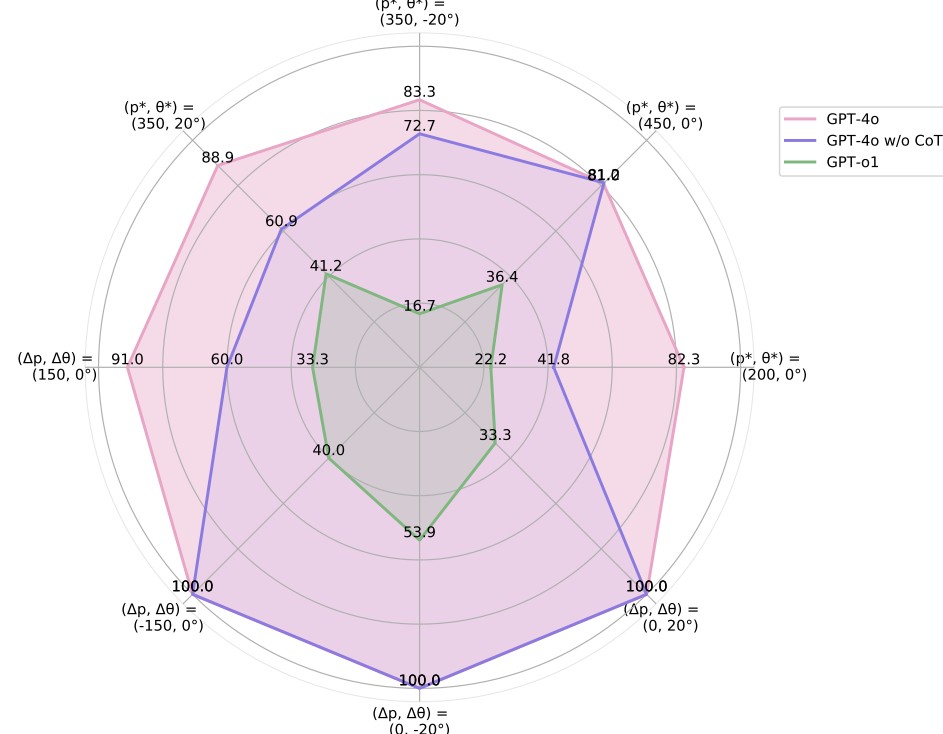

Figure 8: The accuracy of generated bounding boxed based on textual instructions and spatial goals by using GPT-4o, GPT-4o w/o CoT, and GPT-o1.

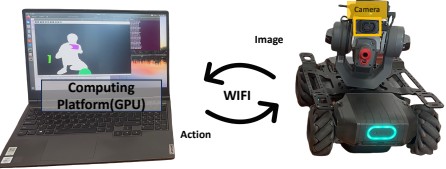

Figure 9: In this real-world deployment, the robot's onboard camera captures visual data, which is wirelessly transmitted to a laptop for real-time processing. The model on the laptop interprets the incoming images, generates appropriate action commands, and transmits them back to the robot via WiFi, enabling precise control of its movements.

the robot's real-time observations. Comprehensive video demonstrations of these experiments are available on our project website (https://sites.google.com/view/hievt).

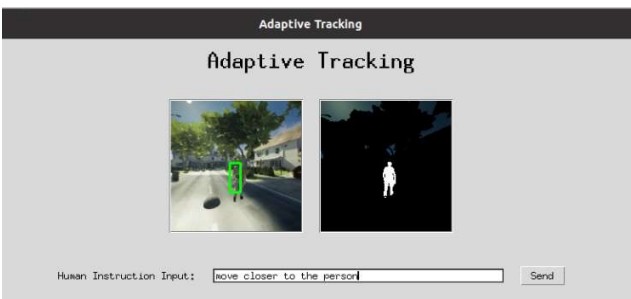

Figure 10: A real-time interactive GUI snapshot. The left panel displays the tracker's first-person view, while the right panel shows the text-conditioned mask generated by the game engine. Users can input instructions in the text box at the bottom or draw a bounding box (e.g., the green box in the left image) and click the "Send" button to interact with the system.

## J.1 HARDWARE SETUP

To evaluate our proposed method in real-world scenarios, we use RoboMaster EP [1] a 4-wheeled robot manufactured by DJI, as our experimental platform (Figure 9). Equipped with an RGB camera, the RoboMaster captures images and allows direct control of linear speed and angular motion via a Python API. This advanced design ensures precise control over the robot's movements during our experiments. To enable real-time image processing, we use a wireless LAN connection to transmit camera data to a laptop with an Nvidia RTX A3000 GPU, which acts as the computational platform to run the model and predict actions based on raw pixel images. The corresponding control signals are then sent back to the robot, completing a closed-loop control system. This seamless integration of hardware and software allows us to execute complex tasks efficiently and accurately, advancing sim-to-real experimentation. In the training phase, we use a continuous action space, which enables us to directly map the speed to robot control signals. The mapping relationship is shown in Table 13.

## J.2 EXPERIMENTAL RESULTS AND OBSERVATIONS

The real-world experiments were conducted to verify three key hypotheses: (1) the system can maintain robustness across diverse physical scenarios, (2) our hierarchical framework can effectively bridge the gap between language instructions and tracking behavior in real environments, and (3) the intermediate spatial goal representation provides sufficient flexibility for diverse instruction types.

### J.2.1 DYNAMIC MOVEMENT ADAPTATION

The robot successfully maintained tracking while targets performed unpredictable movements including **sudden direction changes** and **varying speeds( from walking to running).** A critical observation from these experiments was that the system's performance directly validated our hierarchical design approach. The instruction reasoning process (running at approximately 1-2 Hz) did not interfere with the high-frequency control policy, allowing the tracking policy to operate continuously even during instruction processing. This asynchronous execution enabled the system to maintain responsive tracking while simultaneously processing complex instructions—a capability that would be impossible with end-to-end approaches like GPT-4o or OpenVLA where inference latency directly impacts control frequency. Our video demonstrations on project website clearly show this advantage in action: even when new instructions are being processed (visible through UI indicators in the videos), the robot maintains smooth, continuous tracking without pauses or hesitation. This confirms that our hierarchical framework effectively decouples semantic understanding from real-time control, enabling robust performance in dynamic real-world scenarios.

### J.2.2 SPATIAL GOAL ADAPTATION

These visual recordings confirm that our intermediate spatial goal representation effectively bridges the gap between natural language instructions and robot behavior. Our adaptive policy could respond to dynamically changed spatial goals fast and accurately.

---

[1]https://www.dji-robomaster.com/robomaster-ep.html

**System Prompt**

```
Objective:
You are an intelligent tracking agent designed to follow human instructions and
    dynamically adjust your tracking goal between you and the target. Your task is
    twofold:
1.Identify the target category mentioned in the instruction (e.g., "person," "vehicle,"
    "object").
2.Understand the human instruction to determine the tracking goal. The goal is
    represented as an expected bounding box position in your field of view, with the
    center at [cx, cy], width w, and height h. This will guide your tracking strategy
    to align the target object with the specified bounding box. The local control
    strategy will then use this expected bounding box to achieve different tracking
    angles and distances based on human instruction.

Representation detail:
All positions in the task should be represented as normalized bounding box coordinates
    relative to the image size in the field of view with width and height (e.g., [cx,
    cy, w, h]). 'cx' and 'cy' represent the center of the bounding box, and 'w' and 'h
    ' represent the width and height of the bounding box, respectively, all normalized
     to the range [0, 1].

Task Understanding:
1. **Instruction:** A natural language command describing the desired change in the
    tracking of the target (e.g., "Get closer to the person," "Move further from the
    car," "Keep the dog in the center," or "Keep the object on the left").
2. **Current bounding box:** The current bounding box coordinates and size of the
    target in your field of view relative to the image size, normalized to [0, 1] (e.g
    ., "Target position: [cx, cy, w, h]").

Task Definition:
Your task is to:
Extract the target category from the instruction (e.g., "person," "car," "dog") and
    determine the expected bounding box position and size within your field of view
    based on the human instruction and the current target position.

This should include:
1.**Target category:** Based on the human instruction, provide the target category name
    .
2.**Bounding Box Increment:** Based on the human instruction, provide the change in the
     bounding box. This should be represented as [Δcx, Δcy, Δw, Δh], where Δcx and
    Δcy are the changes in the center coordinates, and Δw and Δh are the changes in
    the width and height of the bounding box.

Instructions: Given the provided human instruction, and current position, think step by
     step, decide the target category name and best bounding box increment to meet the
     human's instructions.

Strategy Considerations:
The given Current Position represents the current target distance and angle relative to
     the tracker.
The human instruction represented the demand for expected tracking distance and angle
    between the target and tracker.
You should consider the human instruction first, and transform the abstract instruction
     into the concrete bounding box position increment.
The increment value should be 20% each time with respect to the original proportion of
    the target bounding box.
The increment value should consider the first-person view perspective effect.

Provide the target category name format in "**Target category:** [target category]" and
     the chain-of-though process of the increment of bounding box position and size
    end with the format "**Bounding Box Increment:** chain-of-thought process.[Δcx,
    Δcy, Δw, Δh]" in [output:]

Example:
Example:
[input:]
Instruction: "Get closer to the person."
Current bounding box: [0.51, 0.57, 0.07, 0.14].

[output:]
**Target category:** [person]
**Bounding Box Increment:** The current bounding box indicates it is positioned near
    the center of view. If the instruction wants to get closer to the target, the
    bounding box size should be larger without horizontal change and a slight
    increment in vertical position, which should be increased Δw, Δh and Δcy.[0.0,
    0.05, 0.03, 0.17].
```

Figure 11: System prompt used in instruction parser module.

**System Prompt**

```
You are an intelligent tracking agent designed to generate discrete control actions to
    control the robot to follow the target object at an expected distance and angle.

Representation detail:
Bounding box: The bounding box position and size within your field of view, represented
    as [cx, cy, w, h]. 'cx' and 'cy' represent the center of the bounding box, and 'w
    ' and 'h' represent the width and height of the bounding box, respectively.

Control Actions:  The control actions are discrete and include the following:
    -move forward: control the robot to move forward for 0.5 meters.
    -move backward: control the robot to move backward for 0.5 meters.
    -stop: control the robot to stop moving.
    -turn left: control the robot to move forward for 0.25 meters and turn left for 15
        degrees.
    -turn right: control the robot to move forward for 0.25 meters and turn right for
        15 degrees.

Task Understanding:
1. **Goal bounding box:** This is provided by the user to indicate the expected
    distance and angle between the target and the tracker, which is a bounding box
    format, the agent should try to align the target bounding box with the goal
    bounding box as much as possible.
2. **Target bounding box:** This is provided by user to indicate the current target
    position in the image, in the bounding box format.

Task Definition:
Your task is to give a suitable action from Control actions, and try to align the Goal
    bounding box with the target bounding box as much as possible.
1. **Actions:** Based on the given Goal bounding box and Target bounding box, you
    should provide the best control action from the control actions to align the
    target bounding box with the goal bounding box as much as possible.

Strategy Considerations:
The target bounding box size in the image represents the spatial distance, the smaller
    size corresponds to a larger distance between the robot to the target.
If the center of the target bounding box is on the right side of the goal bounding box
    center, the robot should turn right to align the target bounding box with the goal
    bounding box. In contrast, if the center of the target bounding box is on the
    left side of the goal bounding box center, the robot should turn left to align the
    target bounding box with the goal bounding box.

Instructions: Given the provided target bounding box and goal bounding box, decide the
    best action to adjust the robot's position.
Provide ONLY and the increment of bounding box position and size in [output:] format in
    [Control Action] without additional explanations or additional text.
```

Figure 12: System prompt used in baseline method GPT4-o.

