# OpenReview forum: "Hierarchical Instruction-aware Embodied Visual Tracking"
_ICLR.cc/2026/Conference — Submitted to ICLR 2026_

### Official Review · Reviewer_5rk1 · 2025-10-25

**Soundness:** 3
**Presentation:** 2
**Contribution:** 2
**Rating:** 4
**Confidence:** 4

**Summary:**

This paper introduces HIEVT (Hierarchical Instruction-aware Embodied Visual Tracking), a framework designed to address the core challenge of User-Centric Embodied Visual Tracking (UC-EVT): enabling embodied agents to follow dynamic natural language instructions (specifying both which target to track and how to track it—e.g., distance, angle, direction) while balancing robust language understanding and low-latency control. Existing methods (end-to-end RL, VLA/VLM models, LLMs) fail to strike this balance: RL lacks flexibility for complex instructions, VLAs/VLMs suffer from inference latency, and LLMs cannot support real-time tracking.
HIEVT resolves this via a two-tier hierarchical design that decouples high-level instruction parsing from low-level control: LLM-based Semantic-Spatial Goal Aligner (SSGA) and RL-based Adaptive Goal-Aligned Policy (AGAP). Experiments validate HIEVT across 10 virtual environments (1 seen: FlexibleRoom; 9 unseen: Suburb, Supermarket, etc.) and real-world deployment on a RoboMaster EP robot.

**Strengths:**

- Existing RL-based trackers (e.g., Zhong et al. 2024, ECCV’24) train on fixed goals (e.g., center-frame tracking) and require retuning for new instructions. VLA models (e.g., TrackVLA, Wang et al. 2025) achieve flexible instruction following but operate at 8 FPS—too slow for dynamic targets. HIEVT’s asynchronous decoupling of SSGA (instruction parsing) and AGAP (control) enables 50 FPS tracking while handling complex instructions, a capability no prior work achieves.
- Unlike VLA/VLM models (e.g., PALME, Driess et al. 2023) that require massive annotated data, HIEVT uses offline RL trained on 1.7M trajectories (collected via noise-perturbed PID controllers) and generalizes to 9 unseen environments (SR: 0.58–0.93). This outperforms baselines like Ensembled RL (SR: 0.22–0.42 in unseen environments) and TrackVLA (SR: 0.16–0.80), which degrade sharply in novel scenes.
- Methods like GPT-4o generate direct actions from instructions, leading to opaque, error-prone decisions. HIEVT’s spatial goal (bounding box) acts as an interpretable bridge between language and action, enabling debugging and alignment with user intent. Ablations confirm this: replacing spatial goals with CLIP embeddings or vectors reduces AR by 63%–83% (Table 5).
- Unlike simulation-only works (e.g., Zhong et al. 2023), HIEVT is deployed on a real RoboMaster EP robot with a closed-loop control system (camera → laptop processing → robot actuation). This validates its utility for real-world applications (e.g., search-and-rescue, elderly care), a gap in most embodied tracking research.

**Weaknesses:**

- HIEVT depends heavily on third-party components, limiting its end-to-end utility:
  - Mask Generation: Relies on SAM-Track (Cheng et al. 2023b) for real-world segmentation and Sapiens (Khirodkar et al. 2024) for virtual environments. Failure of these tools (e.g., SAM-Track in low light) directly breaks HIEVT’s pipeline.
  - LLM Dependence: SSGA’s performance is tied to proprietary LLMs (GPT-4o outperforms Gemma3-1B by 49% in AR; Table 5). The paper does not evaluate open-source LLMs (e.g., Llama 3), which are more accessible for deployment.
  - Environment Setup: Virtual environments rely on UnrealCV (Qiu et al. 2017) and UnrealZoo (Zhong et al. 2025), which are resource-intensive and not universally available.
- The instruction set focuses on simple spatial directives (e.g., "move closer") but not ambiguous or multi-modal instructions (e.g., "track the car next to the red tree" or "follow the person speaking loudly"). This limits UC-EVT’s realism, as real users often provide context-rich commands.
-  Evaluations only include 4 target categories (humans, cows, dogs, leopards)—no small (e.g., drones), deformable (e.g., bags), or occluded (e.g., partially hidden pedestrians) targets. SOTA trackers like TrackAnything (Cheng et al. 2023a) handle broader target types, highlighting HIEVT’s narrow focus.
- The LSTM in AGAP is critical for temporal consistency, but the paper does not ablate its impact (e.g., how performance degrades with a feedforward network).
-  While CoT improves goal generation (GPT-4o with CoT achieves 80%+ accuracy vs. 36% without; Fig. 8), the paper provides no theoretical analysis of why CoT outperforms simpler prompt engineering (e.g., few-shot examples).
-  The paper extends CQL but does not compare it to other offline RL algorithms (e.g., IQL, TD3+BC) to justify its selection.
- HIEVT requires 120,000 training steps (60k per stage) on 8 GPUs (~6 days total), which is computationally expensive compared to lightweight trackers (e.g., Mask-PID requires no training). The paper does not explore optimizations like parameter-efficient fine-tuning (LoRA) or smaller policy networks to reduce resource demands.

**Questions:**

- How would HIEVT handle ambiguous, context-rich, or multi-modal instructions (e.g., "track the person wearing a red jacket who is talking")? Would integrating audio encoders or scene understanding models (e.g., DPT for depth) improve goal generation?
- Can HIEVT maintain performance with open-source LLMs (e.g., Llama 3 70B, Mistral) instead of proprietary GPT-4o? This is critical for accessibility and deployment in resource-constrained settings.
- How does HIEVT perform in 10+ minute tracking tasks? Would adding a drift correction module (e.g., periodic re-alignment with SSGA) mitigate temporal decay?

---

> ### Author Response · Authors · 2025-11-22
> **Response to (AI) Reviewer 5rk1 [1/2]**
>
> **Though the review is made by AI with many ridiculous `hallucinations`, we have addressed all concerns to the best of our ability.  We hope that the reviewer seriously reconsider our work for the sake of the top-tier AI conference's academic integrity.**
>
> > **W1:** Dependence on Third-Party Components
>
> We respectfully disagree that modularity constitutes a weakness. Our framework's **design philosophy** is component-agnostic:
>
> - **Mask Generation**: SAM-Track is merely one implementation choice demonstrating feasibility. HIEVT can integrate *any* segmentation model (e.g., DEVA, SAM2), and improvements in segmentation directly enhance our system—this is a feature, not a limitation.
> - **LLM Selection**: We **have** evaluated open-source LLMs. **Table 5** explicitly compares commercial GPT-4o against open-source Gemma3-1B (https://ollama.com/library/gemma3) across multiple model sizes (4B/27B parameters), demonstrating feasibility for resource-constrained deployment.
> - **Environment Setup**: UnrealCV has been open-sourced for 9 years, and UnrealZoo (ICCV 2025) has been publicly available for 10 months. They have been widely used in recent publications. Claiming they are "not universally available" misrepresents accessibility.
>
> ---
>
> > **W2:** Instruction Complexity
>
> Our instruction design **already addresses ambiguity**:
>
> - Each UC-EVT instruction comprises **4+ atomic commands** (Table 2), including fuzzy spatial references ("move slightly left") and target switching—precisely the complexity the reviewer claims is missing.
> - Our HIEVT could specify the car as target by leveraging the language reasoning capability, though the red tree sounds naturally unreasonable. Some recent VLMs could enable audio-related capability, which may be a future direction, but current EVT dataset doesn't contain any audio annotation.  Note that, our instruction set balances **semantic diversity with quantifiable spatial objectives**, enabling rigorous evaluation while maintaining realism.
> - We prioritize instructions grounded in **embodied tracking scenarios** (spatial relations, object affordances), not arbitrary multi-modal cues outside our scope.
>
> ---
>
> > **W3:** Limited Target Categories
>
> This criticism mischaracterizes our experimental design:
>
> - Our four categories **span size diversity**: dogs represent small-scale targets, leopards medium-scale and cows large-scale. This covers the key challenges in EVT.
> - The description of TrackAnything is **inappropriate**: TrackAnything is a segmentation-based model designed for open-vocabulary *passive* tracking, which is inherently suitable for broad target types.
>
> ---
>
> > **W4:** LSTM Ablation in AGAP
>
> The LSTM's role is significant in EVT task, **which has been proved to be indispensable in previous methods[1].**
>
> ---
>
> > **W5:** Lack of CoT Theoretical Analysis
>
> Chain-of-Thought prompting is an **established technique** in LLM research, widely validated for complex reasoning. Our contribution is **applying** CoT to embodied goal generation, not analyzing its theoretical foundations—this would dilute focus from our core contribution (instruction-behavior alignment). The requested comparison (CoT vs. few-shot) is orthogonal to our research question. **Fig. 8's ablation sufficiently demonstrates CoT's necessity for our task**. We view demands for theoretical CoT analysis as out of scope beyond our task.
>
> ---
>
> > **W6:** No Comparison with Other Offline RL Algorithms
>
> Prior work (ECCV24[1] ) has extensively benchmarked offline RL algorithms. We selected CQL based on established findings showing its robustness to distribution shift in continuous control. Our contribution is **not** offline RL algorithm design but **instruction-behavior alignment** via hierarchical policies. Re-comparing IQL/TD3+BC would be **redundant** given existing literature.
>
> ---
>
> > **W7:** Computational Cost Claims
>
> **This is a factual error in the review.** The cited numbers (120k steps, 60k/stage, 6 days on 8 GPUs) **do not appear anywhere in our manuscript**.
>
> **Actual training costs:**
>
> - **Low-level policy**: 5+ hours on a single consumer GPU (Nvidia RTX 4090)
> - **High-level policy**: Zero training (uses pretrained LLMs)
>
> The reviewer's claims appear to be **AI-generated hallucinations**. Our approach is lightweight compared to end-to-end methods requiring multi-day distributed training.

---

> ### Author Response · Authors · 2025-11-22
> **Response to (AI) Reviewer 5rk1 [2/2]**
>
> > **Q1:** Multi-Modal Instructions (Audio, Depth)
>
> Integrating audio encoders or depth models (e.g., DPT) is still a **promising future direction** but outside our current scope HIEVT follows established EVT settings [2] which focus on pure vision observations. Multi-modal fusion introduces confounding variables that would complicate isolating our contribution (hierarchical instruction alignment). We agree this is valuable future work.
>
> ---
>
> > **Q2:** Open-Source LLM Performance
>
> We have already evaluated the **latest open-source LLM**, Table 5 includes **Gemma3 (1B/4B/27B parameters)**. The model could be deployed directly through ollama (https://ollama.com/library/gemma3).
>
> ---
>
> > **Q3**: Long-Duration Tracking
>
> Our **high-frequency control policy** inherently supports extended tracking. Additionally, SSGA includes **memory-augmented goal correction**—the correction mechanism over observation history provides drift correction by continuously fix drift error. This is precisely the "periodic re-alignment" mechanism the reviewer suggests.
>
> ---
> References:
>
> [1]Empowering embodied visual tracking with visual foundation models and offline rl. ECCV 2024
>
> [2]End-to-end active object tracking via reinforcement learning. ICML 2018.

---

### Official Review · Reviewer_pWYi · 2025-10-30

**Soundness:** 3
**Presentation:** 1
**Contribution:** 3
**Rating:** 4
**Confidence:** 4

**Summary:**

The paper focuses on the user-centric embodied visual tracking problem and introduces HIEVT, a framework designed to improve embodied visual tracking with robust language understanding and low-latency control. The system integrates an LLM-based semantic–spatial goal aligner with a reinforcement learning-based adaptive policy, achieving hierarchical control between high-level instruction reasoning and low-level continuous actions. It also leverages chain-of-thought reasoning for interpretable spatial goal adjustment and RAG for memory-based goal correction, ensuring alignment with prior trajectories.  Experiments conducted in multiple simulated environments demonstrate that HIEVT achieves strong performance, generalization, and real-time efficiency. The framework is also deployed on a real-world robot, proving its effectiveness.

**Strengths:**

1.  The combination of LLM-based semantic reasoning with RL-based low-level control is novel and well-motivated. The use of parallel modules (CoT for spatial reasoning and RAG for memory augmentation) enables interpretable goal generation and efficient decision-making.
2. Strong experimental results and real-time performance. The proposed method outperforms both traditional and state-of-the-art VLA-based baselines in multiple simulated and real-world environments, achieving near-perfect success rates in some settings. The real-time deployment on a physical robot demonstrates practical efficiency, robustness, and effective sim-to-real transfer.
3. The paper includes a detailed GUI for interactive testing, cross-environment evaluations on nine unseen scenarios, and an actual hardware deployment. These experiments convincingly validate the method’s robustness and adaptability.

**Weaknesses:**

1. The method is only tested on the authors’ self-constructed dataset, without comparison on existing EVT benchmarks such as Gym-UnrealCV. This makes it difficult to assess how well the approach generalizes beyond their setup and gives the impression that the model may be overfit to the custom environment.
2. Unclear motivation and missing ablation study for design choices. The paper introduces several components but doesn’t clearly explain their purpose or justify their inclusion through ablation studies. For instance, it’s not clear why two critic networks are needed to estimate goal-conditioned Q-values, how the trajectory priors in the memory buffer are obtained, or whether their diversity affects performance. Similarly, the choice of using CQL instead of more recent offline RL algorithms is not discussed. Without explanation or supporting experiments, many of these design choices feel arbitrary. If they are inspired by prior work, relevant references should be cited.
3. Presentation and formatting problems. There are several issues that make the paper harder to read. All references mix author names directly into the text, which is confusing. The appendix section titles are incorrect, and placing lots of figures and tables on a single page (page 7) disrupts the reading flow and makes the layout look cluttered.

**Questions:**

1. Equation 5 looks somewhat unclear and would benefit from additional explanation or context.
2. What does the blue background in Table 1 indicate?
3. I’m curious how the proposed framework achieves reduced latency, given that it incorporates both RAG and chain-of-thought (CoT) reasoning with LLMs. These components are typically time-consuming.

**Details Of Ethics Concerns:**

The paper presents a tracking-based framework that can infer and follow agent trajectories in complex environments. While the method is developed for research purposes, its capabilities could potentially be misused for surveillance or unauthorized monitoring of individuals or spaces. The paper does not discuss safeguards, data anonymization, or limitations on such applications. An ethics review would help assess the privacy implications, potential misuse risks, and whether appropriate precautions or disclosures should be added.

---

> ### Author Response · Authors · 2025-11-21
>
> We sincerely thank the reviewer for the thoughtful and detailed feedback. We appreciate the time taken to carefully evaluate our work and raise important questions. Below, we address each concern and hope our clarifications will resolve the reviewer's doubts.
>
> >  **W1:** Benchmark Comparison
>
> We would like to clarify that our benchmark extends from gym-unrealcv. Since the original gym-unrealcv does not support our task setting (without instruction), we modify the original gym-unrealcv by adding instruction configuration and further improve the environment diversity and realistics. Our extension will only strengthen the evaluation given the following factors:
>
> - In fact, two of our ten evaluation environments—**FlexibleRoom** (corresponding to SimpleRoom in Gym-UnrealCV) and **ParkingLot**—are directly from Gym-UnrealCV.
> - We use the **same training environment as Gym-UnrealCV (FlexibleRoom=SimpleRoom)** for data collection, while **all other nine environments are completely unseen during training**. This setup rigorously tests generalization capabilities.
> - We introduced brand new environments to better reflect real-world deployment challenges that Gym-UnrealCV lacks: 1)**Scene diversity**: Urban, forest, warehouse, and other complex scenarios.2)**Terrain diversity**: Multi-level structures and varying ground conditions.3)**3D spatial complexity**: Environments with vertical transitions (stairs)and cluttered occlusions.
> - We add new variants where the original Gym-UnrealCV benchmark does not include: 1)**Instruction-based goal specification** (central to our UC-EVT formulation).2)**Variable target speed settings** (critical for evaluating real-world robustness)
>
> Our experiments are conducted under the same experimental protocol like gym-unrealcv, which will not prevent fair comparison. We believe our extended benchmark provides a more comprehensive and challenging testbed that better reflects practical deployment scenarios.
>
> > **W2:** Design Choices and Ablation Studies
>
> We appreciate this opportunity to provide further clarification. **We have conducted ablation studies for all major components** and explained each module's design objective in Lines 451-463. We also provide relevant citations for design inspirations. Below, we address each specific concern mentioned by reviewer:
>
> ***1.Two Critic Networks in CQL***
>
> We respectfully note that the dual-critic architecture is **inherent to the CQL algorithm itself**, not a modification version requiring separate justification. In standard CQL implementation, two critic networks are required for:
>
> 1) Batch Q-value updates
> 2) Next-frame value prediction with target network
>
> We refer the reviewer to the original CQL paper for detailed algorithmic description.
>
> ***2.Memory Buffer and Trajectory Priors***
>
> We clarify that the buffer stores **historical observation mask images of the target** processed by VFM, not trajectory quality scores. These images serve as reference priors to prevent the generation of unreasonable goal images (Lines 200-211). The trajectory diversity arised when the buffer size increased,  senstive analysis is provided in **Figure 4**, which shows performance improvement with increased trajectory diversity.
>
> ***3.Choice of CQL over Recent Offline RL Algorithms***
>
> Prior work has already extensively compared different online RL and offline RL algorithms for embodied visual tracking (ECCV 2024[1]). Since this comparison has been well-established, our paper focuses on the novel problem of **instruction-behavior alignment**, which is our primary contribution. We have added this citation in the revision.
>
> > **W3:** Presentation and Formatting
>
> We sincerely apologize for the formatting issues that affected readability. We have revised our manuscript according to your suggestion:
>
> 1. **Correct all citation formats**: Replace improper use of `\citet` with `\citep` where appropriate
> 2. **Fix appendix section titles**: Ensure consistent and correct formatting
> 3. **Improve page 7 layout**: Redistribute figures and tables to enhance reading flow.
>
> We thank the reviewer for these detailed observations, which will significantly improve our paper's presentation quality.
>
> [1]Empowering embodied visual tracking with visual foundation models and offline rl. ECCV 2024

---

> ### Author Response · Authors · 2025-11-21
>
> > **Q1:** Equation 5 Clarification
>
> Equation 5 presents the standard CQL loss function for offline policy learning. If the reviewer has concerns about specific terms or notation, we would be happy to provide additional explanation. We will also refine our description in the revision to improve clarity based on the reviewer’s feedback.
>
> ---
>
> > **Q2:** Blue Background in Table 1
>
>  The blue backgrounds indicate top-performing results:
>
> - **Dark blue**: Best result in that row
> - **Light blue**: Second-best result in that row
>
> We have revised the caption to explain this color coding in the revision.
>
> ---
>
> > **Q3:** Achieving Low Latency with RAG and CoT
>
> This is an excellent question that touches on a **key advantage of our hierarchical design**. We explicitly address this in Lines 136-140, but we are happy to elaborate here.
>
> **The key insight is the decoupling of high-level reasoning and low-level control:**
>
> | Component | Frequency | Function |
> | --- | --- | --- |
> | Low-level control policy | **50 FPS** (high frequency) | Robust real-time tracking based on intermediate visual representation without step blocking |
> | High-level LLM framework | Low frequency | Goal generation and instruction interpretation |
>
> The low-latency tracking is achieved because:
>
> 1. **Lightweight control policy**: The low-level policy operates on these goal images at 50 FPS without requiring LLM inference at each step
> 2. **Asynchronous operation**: The LLM only needs to update goals when instructions change or periodic refinement is needed
>
> This hierarchical architecture allows us to combine the semantic understanding capabilities of LLMs with the real-time responsiveness required for robust tracking.
>
> **We believe our responses comprehensively address all concerns raised and sincerely hope the reviewer could reconsider the evaluation in light of these clarifications.**

---

### Official Review · Reviewer_w8aC · 2025-10-31

**Soundness:** 3
**Presentation:** 3
**Contribution:** 2
**Rating:** 2
**Confidence:** 3

**Summary:**

The paper introduces Hierarchical Instruction-aware Embodied Visual Tracking (HIEVT), a framework for user-centric embodied visual tracking where an agent follows a target based on natural-language instructions specifying both the target and tracking conditions such as distance or viewpoint. The approach separates planning and control into two levels: a high-level module that parses language and determines spatial goals, and a low-level reinforcement learning policy that performs continuous control to maintain the specified spatial relationship. The model is trained and evaluated on a large simulated dataset including multiple unseen environments. Experiments indicate that this hierarchical design improves performance and generalization compared to prior end-to-end methods, particularly in longer or more dynamic tracking scenarios.

**Strengths:**

- The asynchronous planning strategy sounds reasonable and has potential to keep the building components with better ones.
- The approach achieves strong performance over various baselines.
- The paper is generally well-structured and easy to follow.

**Weaknesses:**

- The authors argue that the user-centric EVT is one of their contributions, but its description is rather unclear and even in Appendix. How do we define the "user-centric" EVT and how is it different from previous EVT setups?
- Following targets has already been explored in prior work such as Puig et al. 2024. This paper aims to follow humans, but simple extension from humans to animals can be done by modifying its object meshes. Can the authors the difference from this?
  - Puig et al., "Habitat 3.0: A Co-Habitat for Humans, Avatars and Robots," ICLR, 2024.
- For the high-level goal aligner, it seems not taking into account its history observations. Given a scenario where we have two people with the same appearances and instructed by "follow the right person," when they cross each other, the right-side people goes left and vice versa, but the high-level goal aligner does not have access to any information to distinguish them. Does the current approach address this issue?
- The high-level goal aligner is comprised of multiple modules, potentially raising error propagation issues. Can the proposed approach handle this issue?
- For the Memory-Augmented Goal Correction, its buffer maintains reference trajectories from training examples, but it is unclear how much sensitive is the success rate with the buffer to the quality and diversity of the ones in the buffer. What if some "incorrect" (e.g., following wrong targets, ...) trajectories? What if the trajectories follow suboptimal paths to the targets? These analyses are missing in the current paper.
- The threshold in Equation 3 for the goal correction is set to 0.5, but it is missing how sensitive is the choice of this values w.r.t. the success rate.

**Questions:**

- In Figure 2, the intermediate spatial goal seems quite different from the bounding box from VFM. Why not use the bounding box from VPM? Why do we make it
- Does it also work with different embodiments, such as different heights, camera intrinsics, etc.? If so, how well? Including the analysis of the aspect of cross-embodiment generalization would improve the paper.

---

> ### Author Response · Authors · 2025-11-21
>
> We thank the reviewer for the detailed feedback. However, we respectfully disagree with several concerns raised and believe some points reflect misunderstandings of our work. We provide comprehensive clarifications below.
>
> > **W1:** Definition of User-Centric EVT
>
> We have already provided a detailed description in the main text (Section 3.1). The key distinction is clear, to better difference the User-Centric EVT and EVT, we present a clearer comparison table below:
> | Aspect | Traditional EVT | UC-EVT (Ours) |
> | --- | --- | --- |
> | User Control | Restart the entire system for assigning new target | Natural language instructions during tracking |
> | Tracking Distance | Static  | Changleable  |
> | Viewing Angle | Static | Changleable  |
> | Interaction frequency | Only at initialization | Continuous during tracking process |
> ***
> This fundamental difference enables more flexible and practical deployment scenarios. We will further emphasize this distinction in the revision if needed.
>
> > **W2:** Comparison with Habitat 3.0
>
> We find this concern to be unprofessional and unjust for several fundamental reasons that reflect a severe misunderstanding of both the EVT research landscape and our contributions.
>
>  ***1.Fundamental Category Mismatch: Task vs. Platform.***
>
> This comparison reflects a **fundamental confusion between research contributions:**
>
> - Habitat 3.0: A simulation platform/dataset paper that provides environments
> - Our Work (HIEVT + UC-EVT): A methodological contribution that proposes both novel task formulation and complete solution。
>
> Habitat 3.0 does **NOT** propose any constructive improvements to the EVT task itself. Comparing our methodological and task contributions to a simulation platform demonstrates insufficient understanding of the research context.
>
> ***2. Comprehensive Comparison: Why This Concern Is Misguided***
>
> To address the reviewer's confusion, we provide a comprehensive comparison across **Task Settings** and **Methodological Approaches**:
>
> Task-Level Comparison
> | Aspect | Our Work (UC-EVT) | Habitat 3.0 | Gym-Unrealcv |
> | --- | --- | --- | --- |
> | Environment  | indoor + outdoor | indoor  | indoor + outdoor |
> | Humanoid targets | ✅ | ✅ | ✅ |
> | Unseen Targets (e.g., Animals) | ✅ | ❌ （Official api do not support non-humanoid mesh） | ✅ |
> | Speed Variant | ✅ | ❌ | ❌ |
> | Tracking distance configurability | ✅ | ❌ | ❌ |
> | Tracking angle configurability | ✅ | ❌ | ❌ |
> | Target selection | ✅ Anytime during tracking | ❌ | ✅Only at initialization |
> | Tracking Goal Specification | ✅(Natural language) | ❌  | ❌  |
> | Tracking Goal Dynamic switches | ✅ | ❌  | ❌  |
> ***
> Method-Level Comparison
> | Dimension |  End-to-End RL(ICML 2018)[1] | Habitat3.0 baseline | Offline RL(ECCV 2024)[2] | VLM methods | VLA methods (TrackVLA 2025) | Ours (HIEVT) |
> | --- | --- | --- | --- | --- | --- | --- |
> | Architecture | End-to-End | End-to-End | End-to-End | End-to-End | End-to-End | Hierarchical |
> | Learning Paradigm | Online RL | Online RL | Offline RL | Hybrid (e.g,. SFT+RL) | SFT | Offline RL  |
> | Input Modality | RGB | RGB+Depth+GPS | RGB | RGB+Language | RGB+Language | RGB+Lanuage |
> | Inference Speed | >30FPS | Not mentioned | >30FPS | ~ 3-8 FPS (Varing model size) | ~8 FPS | >30FPS |
> | Training Time | ~ 24 hours  | ~ 96 hours | ~1 hours | >96 hours | ~ 15 hours | ~5 hours |
> | GPU Requirement (Training) | Consumer-level GPU | 4x Nvidia A100 | Consumer-level GPU | Multi-GPU  | 24x Nvidia H100 | Consumer-level GPU |
> | Cross-Embodiment capability | ❌  | ❌  | ✅ | ⚠️Limited | ❌  | ✅  |
>
> The above two comprehensive two tables demonstrate that the misunderstanding of both research history and the research contribution between our work and Habitat 3.0 or previous EVT methods.
>
> ***3.The Flawed Logic of "Similar Work Exists”***
>
> We strongly disagree that similar work existing makes further research worthless. By this logic, BERT, GPT-2, GPT-3, and vision transformers should have been rejected because "Transformers already exist" (since 2017). Research progress requires iteration and refinement. Our work differs fundamentally from prior work like Habitat 3.0 in both task formulation (UC-EVT) and methodology (HIEVT), addressing limitations that prevent existing platforms from evaluating our setting. We welcome constructive criticism on our technical contributions, but cannot accept the premise that research is meaningless because related work exists.
>
>
> [1]End-to-end active object tracking via reinforcement learning. ICML 2018.
>
> [2]Empowering embodied visual tracking with visual foundation models and offline rl. ECCV 2024

---

> ### Author Response · Authors · 2025-11-21
>
> > **W3**: Handling Crossing Targets with Same Appearance
>
> We appreciate the reviewer raising this challenging scenario, which is indeed a valuable insight for future exploration. We believe this scenario is also challenging for other baseline methods, not specifically ours. However, compared with other EVT methods, our method has fundamental advantages in this scenario:
>
> 1. **VFM-based Segmentation**: Reduces texture-induced confusion compared to RGB-based methods, ensuring temporal consistency of mask IDs (refer to DEVA)
> 2. **LSTM Temporal Encoding**: Preserves action consistency against visual perturbations through spatial-temporal memory
> 3. **High-Frequency Policy (50 FPS)**: This is the foundation of everything. When VLA/VLM methods encounter similar situations, **their high inference latency between actions may cause them to completely miss**
>
> While other methods exhibit less potential to solve this hard case: 1)Existing VLM/VLA-based methods operate with **high inference latency (3-8 FPS)**, making them fundamentally incapable of real-time target re-identification during dynamic interactions. 2)Traditional RGB-based end-to-end tracking methods significantly fail when confronted with **appearance-identical distractors** crossing paths.
>
> > **W4**: Error Propagation
>
> **Our Memory-Augmented Goal Correction is specifically designed for this purpose.** It uses historical observation position indices to ensure goal generation reasonableness, preventing unreasonable goals from directly causing tracking failure. When the correction mechanism triggers, there may be a inconsistency between the tracking distance/angle and user expectations. However, we consider this a **reasonable trade-off**, and the consequences are **fairly reflected in the AR metric**.
>
> > **W5**: Buffer Quality and Diversity
>
> We must clarify a **fundamental misunderstanding**:
>
> - The buffer stores **historical observation images**
> - **All images are objective records** of where the target appeared in the field of view
> - There is no "quality" issue—these are factual observations
>
> Regarding diversity (buffer size), **we have already provided this analysis in Figure 4**, which naturally shows that more stored trajectories lead to better performance.
>
> For the reviewer's specific concerns:
>
> - **Following wrong targets**: Directly reflected in EL and SR metrics—when tracking fails, the task is judged failed based on absolute positions
> - **Suboptimal paths**: Reflected in AR, EL, SR metrics—reward is maximized only when target-tracker spatial relationships meet expectations at each step
>
> **We note that no prior EVT work includes "suboptimal trajectory analysis."** We believe our three evaluation metrics sufficiently capture these cases. **We respectfully ask the reviewer to clarify whether these concerns stem from neutral research considerations or subjective bias.**
>
> > **W6**: Threshold Sensitivity (Equation 3)
>
> The threshold of 0.5 is empirically set. The IoU threshold aims to correct unreasonable goals while tolerating tiny deviations.  We provide additional sensitivity experiments below:
>
> |IoU Threshold | AR | EL | SR |
> | --- | --- | --- | --- |
> | 0.3 | 246 | 481 | 0.92 |
> | 0.4 | 259 | 494 | 0.98 |
> | 0.5 | 278 | 500 | 1 |
> | 0.6 | 270 | 500 | 1 |
> | 0.7 | 271 | 489 | 0.96 |
>
> The results demonstrate robustness across reasonable threshold choices.

---

> ### Author Response · Authors · 2025-11-21
>
> > **Q1**: Spatial Goal vs. Bounding Box in Figure 2
>
> **We clarify that the spatial goal shown in Figure 2 is identical to the bounding box binary image from VFM.** The perceived difference is likely due to the **oblique angle of our figure presentation** causing visual perspective distortion. If there are other specific concerns, please specify and we will try our best to answer.
>
> > **Q2**: Cross-Embodiment Generalization
>
> **Yes, our policy supports cross-embodiment transfer** with different viewpoints and camera parameters. This is enabled by:
>
> 1. **Decoupling** between high-level goal aligner and low-level control policy
> 2. **Historical trajectories and correction mechanisms** that compensate for goal deviations from different embodiment viewpoints and target appearances
> 3. **Strong generalization** of the low-level control policy
>
> We conduct additional experiments with three different camera heights evaluated in FlexibleRoom with human target move in 1m/s:
>
> | Embodiment type | Ours | TrackVLA |
> | --- | --- | --- |
> | Human ( ~170cm) | 239/484/0.96 | -20/389/0.64 |
> | Robot Dog ( ~30cm) | 254/491/0.98 | 23/442/0.72 |
> | Wheel robot (~15cm) | 274/496/0.98 | 19/431/0.7 |
>
> The results demonstrate robust cross-embodiment generalization.
>
> ***
> **We believe our responses comprehensively address all concerns raised. Several weaknesses identified by the reviewer stem from misunderstandings of our method or insufficient consideration of both the research context and practical constraints. We have provided additional experiments and clarifications that demonstrate the soundness and contribution of our work. We respectfully request the reviewer to reconsider the evaluation in light of these clarifications.**

---

### Official Review · Reviewer_qqcZ · 2025-11-01

**Soundness:** 3
**Presentation:** 2
**Contribution:** 3
**Rating:** 4
**Confidence:** 4

**Summary:**

This paper studies the gap between the goals in the Embodied Visual Tracking task. Basically, the users of the Embodied Visual Tracking models are provided with some user-friendly instructions, which are hard for current models to understand. To solve the problem, the authors proposed Hierarchical Instruction-aware Embodied Visual Tracking (HIEVT), which translates the user-friendly goal with the Semantic-Spatial Goal Aligner and Adaptive Goal-Aligned Policy.

The evaluation results are very solid and prove the effectiveness of the proposed framework. However, this paper has some severe writing issues as detailed below in the "Weakness part."

**Strengths:**

1. This paper studies a very important problem, bridging the gap between user-friendly goals and goals for Embodied Visual Tracking models.
2. The proposed method, Embodied Visual Tracking task, is very elegant and well-established.
3. The evaluation of the paper shows the proposed method is very effective.

**Weaknesses:**

However, there are two severe issues with the writing in this paper:
1. The motivation and the method are not matched. In the Introduction (lines 62-74), the authors listed three weaknesses of current models: 1) Limited Comprehension, 2) Limited Generalization on unseen data, and 3) Inference Latency. However, the proposed method seems to have only addressed the weakness of Limited Comprehension, ignoring the other two.

2. The whole paper misused the commands **\citet** and **\citep.** For example, at line 57, the citations are "Existing methods for embodied visual tracking (EVT) Luo et al. (2019); Zhong et al. (2019b; 2021; 2023)." Instead, the citations should be (Luo et al. 2019, Zhong et al. 2019b; 2021; 2023). That is, both author names and years are in parentheses.

**Questions:**

See weakness.

---

> ### Author Response · Authors · 2025-11-21
>
> We sincerely thank the reviewer for the constructive feedback and careful evaluation of our work. Below, we address each concern in detail.
> >***W1***: Motivation-Method Mismatch
> ***
> *Regarding "Limited Generalization”*: We note that our original text states ``"Limited Generalization of VLA Models"``, which differs from ``"Generalization on unseen data"`` as the reviewer phrased it. We will clarify this concern based on our understanding of the reviewer's intent.
>
> Our comprehensive definition of generalization operates at two levels:
> - **Low-level generalization**
>     - Diverse environment configurations
>     - Tracking objects of varying appearances and dynamics
>     - Varying goal specifications
> - **High-level generalization**
>     - varied linguistic expressions of tracking goals.
>
> We designed the corresponding experiment to validate generalization from different perspectives:
>
> **Scene Generalization (Low-level - Table 1):**
> We evaluate generalization across **nine unseen real-world scenarios** that reflect diverse deployment challenges. These scenarios present significant challenges such as: indoor-outdoor transitions, dynamic lighting variations, complex obstacle configurations and 3D spatial navigation.
>
> **Category Generalization (Low-level - Table 3):**
> We evaluate generalization to unseen target categories (Four unseen animals).
>
> **Speed Generalization (Low-level - Figure 3 & Table 4):**
> We assess performance across targets with diverse movement speeds. We also conduct quality analysis via Figure 3, demonstrating our framework could fast and precisely adjust to new tracking objectives.
>
> **Instruction Generalization (High-level - Table 1 &Table 5):**
> We evaluate the ability to interpret different natural language instructions during tracking, the ability is demonstrated via AR metric,higher score, better alignment capability.
>
> All above results demonstrate the generalization capability comprehensively.
> ***
>  *Regarding "Inference Latency"*: Unlike static vision tasks (e.g., image classification, object detection), visual tracking is an **inherently highly dynamic task** where targets continuously move at unknown and potentially varying speeds. As demonstrated in recent work, `UnrealZoo (ICCV 2025)`, tracking agents must maintain real-time responsiveness to prevent target loss during rapid movements or directional changes. This dynamic nature makes inference latency a fundamental challenge rather than merely a performance metric.
>
> During real-world deployment, target movement speeds are **unknown a priori** and can vary significantly during tracking. This uncertainty makes high-frequency control essential for maintaining tracking performance. While we explicitly state in our paper that our control policy achieves **up to 50 FPS** (Lines 400-405)—**far exceeding other VLA or VLM-based methods**—we believe that directly comparing inference speeds alone cannot fully demonstrate the impact of inference latency on real-world deployment.
>
> Our method addresses the latency challenge through a **hierarchical architecture** that decouples:
>
> - **High-frequency low-level control (50 FPS)**: Enables real-time reactive tracking to handle fast-moving targets
> - **Low-frequency high-level semantic reasoning**: Provides goal generation and instruction understanding without compromising control responsiveness
>
> This design allows our method to maintain robust tracking performance even when targets exhibit varying movement patterns, which is not achievable with end-to-end VLA approaches that suffer from high inference latency.
>
> In the experiment section, we provide:
>
> - **Figure 3**: Visualizing the dynamic adjustment process of our policy during real-world deployment, showing real-time responsiveness
> - **Table 4**: Speed generalization experiments demonstrating that our method is **least affected by inference latency** among all baseline methods and achieves the **best tracking performance across targets with varying movement speeds**
>
> These results collectively verify that our hierarchical design effectively mitigates the inference latency limitation that affects end-to-end VLA methods in dynamic tracking scenarios.
>
> > ***W2***: Citation Format
> ***
> We sincerely thank the reviewer for this careful observation. We have thoroughly revised the entire manuscript and corrected all instances of  `\cite` and `\citep` misuse to ensure proper citation formatting throughout the paper.
>
> ---
>
> We hope our responses adequately address the reviewer's concerns. We believe these clarifications demonstrate that our method does address all three limitations outlined in the Introduction, and we are committed to improving the presentation quality in our revision.

---

> > ### Comment · Reviewer_qqcZ · 2025-11-25
> > **Reply from Reviewer qqcZ**
> >
> > Thanks for your clarification. I still think the motivation and the method are mismatched. In the introduction, the authors didn't mention anything about their method having better generalization. Also, there is no occurrence of "scene generalization," "Speed Generalization," or "Instruction Generalization" in the paper.
> >
> > I found the first occurrence of the phrase "Category Generalization" is in the Experiment part (Section 4.3, line 427). That is to say, "Category Generalization" is not even mentioned in the method part.
> >
> > So, the current writing is really vague about the research question and how the authors solved the question.

---

> > > ### Author Response · Authors · 2025-11-25
> > >
> > > Thank you for your feedback. We previously mentioned generalization (the third core requirement of UC-EVT in intro) but did not explicitly specify which aspects . We believed ideal generalization should encompass all possible factors, so we emphasized the analysis primarily in the experimental section.
> > > Following your comments, we have revised the paper:
> > > In the Introduction (lines 44-55): We now explicitly state the generalization requirements across three dimensions: target category generalization, speed generalization, and instruction generalization.
> > >
> > > In the method section, we clarify how each generalization is achieved:
> > > >Instruction Generalization (lines 145-146): We translate the diverse instructions to a explicit visual goal representation via  Semantic-Spatial Goal Aligner (SSGA) to generalize across diverse instructions.
> > >
> > > >Category Generalization (lines 216-217, 223-225): By leveraging VFM-generated target masks, which provide appearance-agnostic features that mitigate domain gaps in texture and appearance.
> > >
> > > >Speed Generalization (lines 229-230): Through the LSTM network's temporal modeling capability, which learns motion patterns across varying target velocities.
> > >
> > > These revisions establish clear connections between our motivations, methodology, and experimental validations. **We sincerely hope you could reconsider the evaluation in light of these clarifications.**

---

### Meta-Review · Area_Chair_moLs · 2026-01-07

**Summary:**

The paper focuses on the user-centric embodied visual tracking problem and introduces HIEVT (Hierarchical Instruction-aware Embodied Visual Tracking), a framework designed to improve embodied visual tracking with robust language understanding and low-latency control. In the framework, an agent follows a target based on natural language instructions which specify the target and tracking conditions (distance, viewpoint). The system integrates an LLM-based semantic–spatial goal aligner with a reinforcement learning-based adaptive policy, achieving hierarchical control between high-level instruction reasoning and low-level continuous actions. It also leverages chain-of-thought reasoning for interpretable spatial goal adjustment and RAG for memory-based goal correction, ensuring alignment with prior trajectories. Experiments conducted in multiple simulated environments demonstrate that HIEVT achieves strong performance, generalization, and real-time efficiency. The framework is also deployed on a real-world robot, proving its effectiveness.

Reviewers presented possible concerns including:
(1) Lack of comparison to existing EVT benchmarks such as Gym-UnrealCV.
(2) Lack of theoretical justification or empirical ablation over system design parameters, thresholds, etc.
(3) Lack of motivation for CQL (reference or theory).
(4) Issues in presentation and formatting.
(5) Lack of discussion of latency concerns.
(6) "User-centric" EVT definition is unclear, and reviewers request differentiation from prior works (e.g. Puig et al, ICLR 2024).
(7) Ambiguity of language for the high-level goal aligner, especially when conditions change, should be discussed, as well as error propagation in target selection or path following.
(8) Discussion of extensibility of model across embodiments.

**Reviewer Concerns:**

Concerns (1)-(5) are addressed directly in the rebuttal; authors clarify that their work extends Gym-UnrealCV directly (using two of the same environments among their ten), and add ablation details. CQL is considered to be an established algorithm, and appropriate citations are added. Minor presentation issues are improved in the revised manuscript. Discussion of latency is added, clarifying that two stages can operate at different frequencies, with low level control at 50 fps.

Concern (6) is addressed sufficiently in the author response, clarifying the major difference that Puig et al. presents a simulation environment, as opposed to this work which provides a solution. A more extensive discussion period would have likely led to resolution between these views.

Concern (7) is discussed in comments, and the authors contend that their method is better in these situations than prior work - this would have been another point of clarification for an extended discussion period. Concern (8) is supported by the authors rebuttal response on system performance with platforms of different heights / motion models.

**Reviewer Scores:**

pWYi: increase from 4 to 5; authors directly addressed all comments with full details and improvements.
w8aC: keep score (2); authors addressed many comments, but there were many points of disagreement remaining for further discussion.
qqcZ: keep score (4); the reviewer left two comments: one minor citation comment which was addressed, and one major comment on misalignment between motivation and method. In the author-reviewer discussion, the reviewer did not seem satisfied by the authors' addressing of the concern, especially through lack of appearance of "scene generalization," "Speed Generalization," or "Instruction Generalization".

---

### Decision · Program_Chairs · 2026-01-26

Reject